# Divergent subregional information processing in mouse prefrontal cortex during working memory
Alex Sonneborn[1], Lowell Bartlett[1], Randall J. Olson [1], Russell Milton [1] & Atheir I. Abbas [1,2,3] ✉

Working memory (WM) is a critical cognitive function allowing recent information to be temporarily held in mind to inform future action. This process depends on coordination between prefrontal cortex (PFC) subregions and other connected brain areas. However, few studies have examined the degree of functional specialization between these subregions throughout WM using electrophysiological recordings in freely-moving mice. Here we record single-units in three neighboring mouse medial PFC (mPFC) subregions—supplementary motor area (MOs), dorsomedial PFC (dmPFC), and ventromedial (vmPFC)—during a freely-behaving non-match-to-position WM task. The MOs is most active around task phase transitions, when it transiently represents the starting sample location. Dorsomedial PFC contains a stable population code, including persistent sample-location-specific firing during the delay period. Ventromedial PFC responds most strongly to reward-related information during choices. Our results reveal subregionally segregated WM computation in mPFC and motivate more precise consideration of the dynamic neural activity required for WM.

Working memory (WM) is a fundamental cognitive function allowing prior sensorimotor and rule information to be held in mind, manipulated, and protected from interference for future use[1]. This process relies on distributed brain networks performing varying degrees of top-down and bottom-up operations[2,3]. The mammalian prefrontal cortex (PFC) is theorized to be a critical WM node, exerting its influence via extensive reciprocal connections with cortical[4] and subcortical structures[5]. It has been implicated in orchestrating several key aspects of WM, including actively directing and maintaining attention toward salient features of a context, selecting strategies to accomplish goals based on contextual needs, and monitoring the outcome of enacted motor plans to change strategies if necessary[6,7]. Difficulty with any of these functions is prevalent across many human neurological and psychiatric disorders and usually coincides with aberrant activation of PFC-containing networks during WM[8,9]. A deeper examination of PFC activity throughout the different phases of WM is critical to better understand the potential mechanisms underlying diverse types of WM dysfunction.

Over the past decade, mice have become a standard model organism in PFC research due to rapid development of genetic tools permitting more precise targeting of cell-types and brain-wide circuits[10]. Extensive connectomic and genomic mapping has established that mouse PFC can be divided into subregions based on local and long-range projection patterns[11,12] and cytoarchitecture[13,14]. However, attempts at segregating mouse PFC into subregions based on functional processing of WM task

features has yielded surprisingly inconsistent findings[15]. A potential explanation for this variability comes from recent work showing that neural activity subserving goal-directed actions is spread across many brain areas, and the primary locus of control can shift dynamically depending on contextual needs and temporal progress through a task[16–22]. Thus, trying to localize multifaceted mental processes, like WM, onto isolated mouse PFC subregions may be an unreliable approach. Instead, experiments in mice should focus on characterizing a range of WM-related computations in multiple PFC subregions over the entire time course of a single behavioral paradigm[15].

Most mouse studies probing PFC neural circuit contributions to spatial WM have concentrated on single subregions within the same task[23–26], and reports using electrophysiological recordings from multiple subregions are not accompanied by a detailed comparison between them[19,27–29]. Moreover, most multi-regional WM data have been collected from head-fixed mice[30–33]. Critically, more comprehensive analysis describing how distinct PFC areas selectively contribute to WM task variables across time in freely-moving mice is needed, as head-fixed experiments may engage different brain networks than more naturalistic behaviors[34,35]. To this end, we recorded single-units in three adjacent mouse PFC subregions, agreeing with modern PFC parcellation schemes[15]: the supplementary motor cortex (MOs), the dorsomedial PFC (dmPFC), and the ventromedial PFC (vmPFC). Activity in these subpopulations was tracked in real-time as the mice performed a

[1]Department of Behavioral Neuroscience, Oregon Health & Science University, Portland, OR, USA. [2]Department of Psychiatry, Oregon Health & Science University, Portland, OR, USA. [3]Research and Development Service, VA Portland Health Care System, Portland, OR, USA. ✉e-mail: abbasat@ohsu.edu

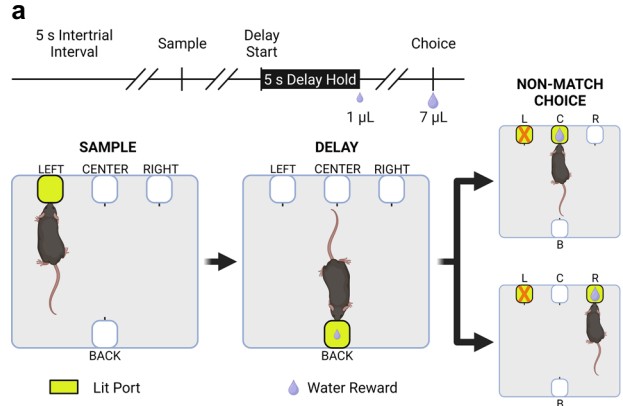

**Fig. 1 | A freely-moving delayed non-match-to-position task allows for the examination of prefrontal neural activity during spatial working memory in mice. a** Schematic of the delayed non-match-to-position task. (top) Time course of a single trial. Diagonal parallel slashes represent variable amounts of time between task phases. (bottom) Diagram of a correct trial. Progression through the task starts with the sample phase (left panel), during which one of the two outer front ports lights up (the center port was never used as a sample location). The lit sample port location has a 50% chance of being on either the left or right side, but only a left side example trial is shown here. The mouse pokes its nose into the lit sample port to make the delay port on the opposite side ("back") of the box available (center panel). Then the mouse pokes and holds its nose in the delay ("back") port for five seconds, after which it receives a 1 μL reward. This leads to a choice phase (right panel), where both the initial sample port and one of the remaining other front ports lights up, and

the mouse is required to poke into the lit port that it has not visited previously. Non-match choices also have a 50% chance of being either of the two non-sample locations on any given trial. A much larger 7 μL reward is dispensed after a correct non-match choice. Created in BioRender. Palumbo, M. (2023) BioRender.com/u81z167. **b** Mice were implanted with recording electrodes in either MOs (blue), dorsal mPFC (green), or ventral mPFC (pink). Example coronal mouse slice showing the final electrode bundle locations after up to five electrode advancements of ~60 μm (white dots). Across all sessions, we isolated 304 single-units from the MOs, 354 from the dmPFC, and 330 from the vmPFC. Brain slice image credit to the Allen Mouse Brain Atlas. **c** A one-way ANOVA showed that electrode location did not significantly affect task performance (F(2,70) = 2.71, p = 0.073, n.s. not significant). Each circle represents performance during one session.

freely-behaving delayed-non-match-to-position WM task[36]. We asked how each subregion represented the retrospective sample location and other task-related information, such as prospective choices and reward, across the encoding, maintenance, retrieval, and outcome phases of WM[37]. Our results indicate that the dmPFC population stably codes for the retrospective sample location throughout all task phases in single behavioral trials, while MOs prioritizes sample identity and other information at contextual transitions and vmPFC is most reflective of choices and their outcomes.

## Results

### Behavior and electrode implantation

To examine the relationship between spatial WM and PFC subregional activity, mice were water restricted to ~90% of their initial body weight and trained on a freely-moving, delayed non-match-to-position (DNMTP) WM task (Fig. 1a). Critically, this task was designed so that mice could not know the exact choice port they would need to visit until the end of the delay period. During training, mice that made the correct non-match choice on ≥70% of trials over three consecutive days were implanted with a custom-built, 28-wire, advanceable bundle of microelectrodes into one of three separate PFC subregions: MOs (blue, four mice), dmPFC (green, six mice), or vmPFC (pink, six mice) (Fig. 1b). After one week of recovery, we gathered at least three daily sessions of simultaneous behavioral and electrophysiological data per mouse, advancing the electrodes ventrally into the PFC by ~60 μm after each session, so that new neurons were recorded the following day (white dots in Fig. 1b depict the final electrode bundle locations for each mouse). Single-units were isolated offline using Kilosort3, aligned to important DNMTP task events, and organized into pseudopopulations by combining the neurons recorded over all sessions within each subregion. The total neuron count for each pseudopopulation was 304 from the MOs (over 21 sessions), 354 from the dmPFC (over 24 sessions), and 330 from the vmPFC (over 28 sessions). Figure 1c quantifies the behavioral performance in all post-implantation sessions in which neurons were recorded and used for analysis. A one-way ANOVA showed no significant main effect of electrode implant location on DNMTP performance (F(2,70) = 2.71, p = 0.073, Fig. 1c).

One possible confound of this task is that the center port always gives a reward when lit, since it is only available as a choice and never used as a sample location (Fig. 1a). Therefore, mice could conceivably develop a strategy where they treat the center choice trials as simple visuospatial stimulus-response trials. If this were the case, they should perform at nearly 100% on center choices[38] (50% of all trials) and would only need to achieve chance level (50% performance) on outer trials (remaining 50% of trials, 25% left and 25% right) to reach ~75% performance overall, which was above the performance cutoff for including a session in our analyses. To check if this was happening, we plotted the trial-type-specific performance across all sessions and saw that mice did perform better on average when the center port was the correct choice (Supplementary Fig. 1a). However, when plotting the correlation between the performance on center choice trials versus the performance on left and right choice trials combined, less than 10% of the sessions (Supplementary Fig. 1b, circled in black) had near 50% outer performance with high center performance, the pattern that would be expected if mice were exploiting the task and prioritizing the center port. The remaining >90% of sessions had a positive correlation between center and outer performance, a pattern more consistent with the mice using a general WM strategy which applies to all the ports. Interestingly, six of the seven sessions with a concerning pattern of performance were from the same dmPFC-implanted mouse (dmPFC 4; see green dots circled in black in Supplementary Fig. 1b), meaning that this mouse was likely not using a WM-based strategy.

We further reasoned that if the mice were using a simple stimulus-response strategy at the center port, they would respond more quickly on the easier center trials than on outer trials. This was not the case, and the time it took the mice to move from the back delay port to a front choice port for center versus outer choice trials was again positively correlated (Supplementary Fig. 1d), suggesting that the mice treated all choice port locations similarly (Supplementary Fig. 1c). Overall, these results point toward the mice (with one exception) using an active WM strategy, and the increased performance on center choice trials could potentially be explained by the mice having twice the amount of training on these trials compared to choice trials for either outer location.

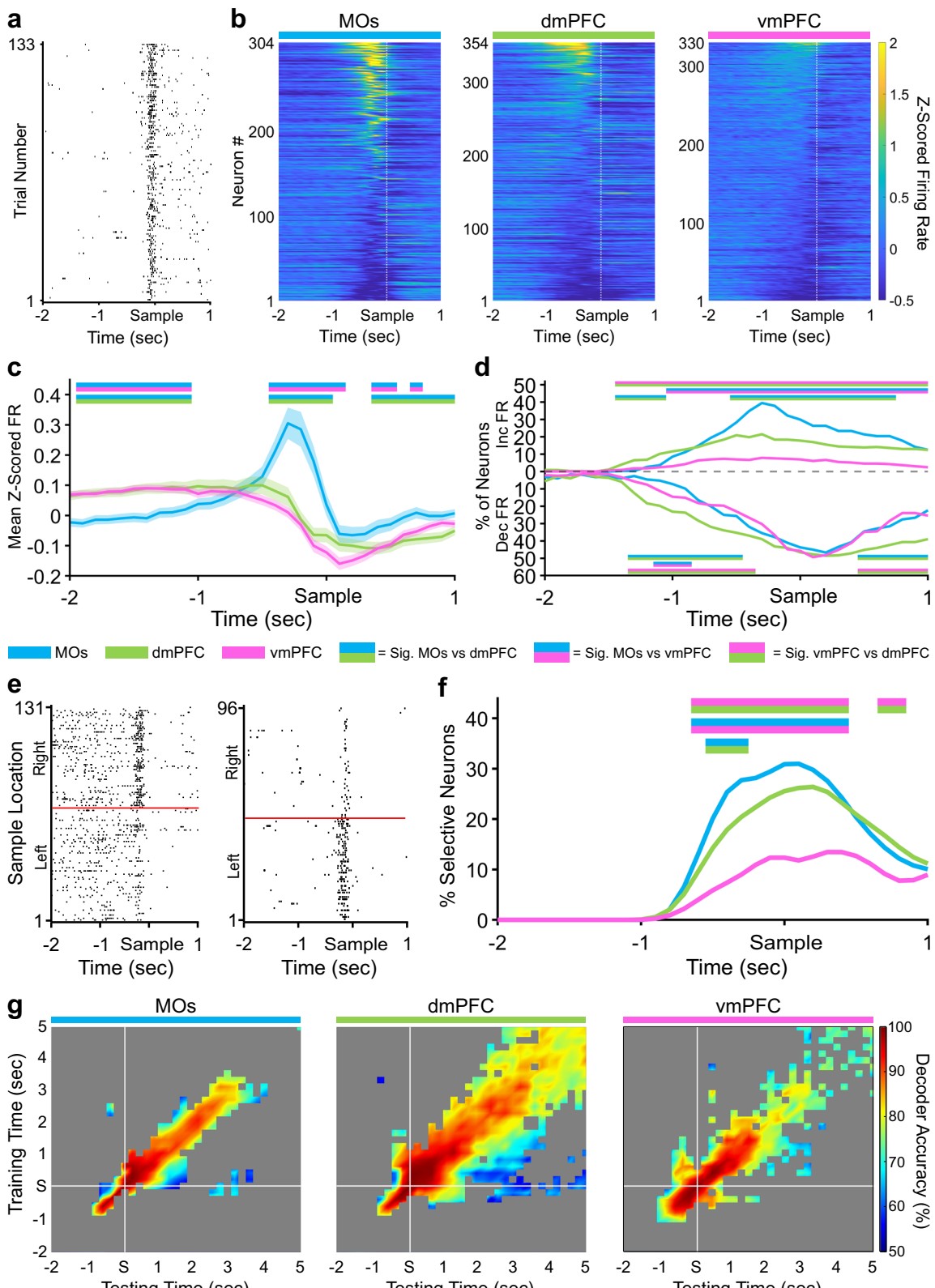

## MOs is more active and contains the most sample identity-selective neurons during the DNMTP sample phase

We next tracked subregional neural firing activity throughout all key phases of the DNMTP task. Each pseudopopulation contained neurons exhibiting task-related changes in firing rate (FR) that consistently appeared within a brief time window around the sample phase nose-poke across many trials

(example neuron in Fig. 2a). Heat maps of the FR were created by first Z-scoring the spike counts in 100 ms bins for all neurons in separate pseudopopulations, followed by sorting them from highest to lowest peak Z-scored activity around the sample nose-poke for better visualization (Fig. 2b). Quantification of mean Z-scored pseudopopulation FR at each time point revealed that the MOs significantly increased its FR compared to

**Fig. 2 | Sample location selectivity is highest in the MOs during the sample phase of the DMNTP task. a** Example raster plot from an MOs neuron increasing its firing rate around the sample port poke on all correct trials. **b** Z-scored heat maps of all neurons recorded from each region around the sample poke (white dashed line) sorted by each neuron's mean Z-scored firing rate 500 ms before the sample poke. **c** The mean Z-scored pseudopopulation firing rate is higher in MOs leading up to the sample poke compared to the other regions. Lighter shaded areas above and below the solid lines represent the standard error of the mean at each time point. **d** The percent of neurons significantly increasing or decreasing their firing rate from baseline (ITI) in at least two 100 ms bins is also higher in the MOs. **e** Example sample location-sorted raster plots of two MOs neurons exhibiting transient selective firing for the right or left sample location, respectively. Red line separates left from right sample trials. **f** MOs contains the highest percentage of neurons selective for either retrospective sample location (Left and Right combined) in a ~1 s window around the sample poke. **g** Sample location decoding accuracy of a linear support vector machine trained and tested on every combination of 200 ms time bins from the sample window. White lines indicate the exact time the mouse poked the sample port. Grey area represents non-significant decoder accuracy compared to a trial identity-shuffled control. In panels **c**, **d**, and **f**, double-colored straight lines represent statistically significant differences (p-value < .05) between the two respective subregions in that time bin, after correcting for both false discovery rate and family-wise error rate.

the other two implant locations in a 300–400 ms window before the sample poke (Fig. 2c, see Methods). There were no significant differences between dmPFC and vmPFC at any time points. We also quantified the proportion of neurons in each pseudopopulation that significantly increased or decreased their Z-scored FR around the sample port visit compared to an intertrial interval (ITI) period (Fig. 2d, see Methods). In correspondence with the above finding, the MOs had the largest proportion of units increasing their FR around the sample poke compared to the two other regions, followed by the dmPFC, while vmPFC contained the lowest. Interestingly, in all regions, many of the neurons significantly changing in one direction also exhibited a significant change in the opposite direction at another time point within the time window analyzed (see Supplementary Fig. 2a for proportions of neurons increasing, decreasing, or both; see Supplementary Fig. 2b for two example neurons that both increase and decrease within the same time window). In Fig. 2c, d, and throughout the rest of the paper, time bins with significant pairwise differences ($p < 0.05$, see Methods) between two mPFC subregions are signified by straight lines above or below the data containing their two respective colors.

We next looked at how selective the pseudopopulations were for the sample port location in the same period around the sample poke (examples of selective MOs neurons in Fig. 2e). Using a permutation testing approach (see Methods), we found that the MOs also contained the most neurons that could significantly differentiate sample port location based on their FR (Fig. 2f). This was followed by dmPFC and lastly by the vmPFC. These results suggest a functional gradient, which is strongest in MOs and weakest in vmPFC, in the extent to which these different subregions encode not only the beginning of the sample phase in general, but also the sample port location around the sample poke event.

The stability of this selectivity was subsequently measured using cross-temporal linear support vector machine (SVM) decoding analysis (Fig. 2g, see Methods) extending out five seconds from the sample poke, which should include the start of the delay period on most trials. In the MOs and vmPFC, the decoder had strong and significant on-diagonal (trained and tested on the same time bin) predictive ability which rapidly decreased to chance levels in about 400 ms for the MOs and 600 to 800 ms in the vmPFC. In these two subregions, strong on-diagonal decoding performance ( ≥ 80%) seemed to stop about three seconds after the sample poke (which is likely when the delay phase starts on average), indicating the absence of a stable code extending into the delay period. In contrast, the dmPFC had significant off-diagonal (trained on one time bin and tested on all others) decoding accuracy for up to several seconds, along with a continuous predictive strength of ≥80% for over five seconds after the sample poke, which should be well into the delay period. This indicates a more robust and temporally stable population code in the dmPFC during this time.

Finally, since a larger number of neurons discriminate between sample ports in the MOs, we wanted to check if we could get better SVM decoding with fewer neurons in this region around the sample poke. Subsampling the neurons from all pseudopopulations into progressively larger subpopulations revealed that on-diagonal MOs decoding accuracy was consistently higher than dmPFC and vmPFC when fewer neurons were used in the model (Supplementary Fig. 3a). Interestingly, the same subsampling technique produced lower off-diagonal decoding in the MOs compared to dmPFC and vmPFC (Supplementary Fig. 3b). Overall, these results imply

that the MOs contains a larger proportion of selective neurons, but they are only very transiently selective around the sample poke (e.g., the example neurons from Fig. 2e which are only selective for ~200 ms), while the other two regions have fewer sample-selective neurons, but their selectivity lasts longer.

## Retrospective sample port information is stably maintained in the dmPFC throughout the delay period

The approaches taken above were next applied to the delay phase of the task. MOs Z-scored population activity around the delay poke was elevated above the other subregions to a degree comparable to the sample phase (Fig. 3a–c). This poke-related difference in pseudopopulation FR between subregions did not persist into the five second delay holding period. Similar to its activity in the sample phase, the vmPFC contained the lowest number of neurons significantly changing their activity throughout the delay (Fig. 3d). Combined with the observation that the vmPFC also had remarkably few sample-selective neurons around the delay poke (Fig. 3f), we conclude that this subregion plays a minimal role during the delay period in our task.

The most striking finding during the delay phase was the presence of neurons in the dmPFC which appeared to stably differentiate retrospective sample port identity (left versus right) throughout the five second holding period (see Fig. 3e for two examples). Examination of sample selectivity across the entire delay revealed a time-dependent transition in the subregion encoding the retrospective sample location most prominently. In the 200 ms around when the mice poked in the back delay port, the MOs had the highest proportion of sample-selective neurons by a small but significant margin over the dmPFC (Fig. 3f). Slightly less than one second into the delay, the dmPFC became the only subregion to contain any neurons with retrospective sample selectivity, an effect which lasted until the end of the delay holding period and resembled a persistent working memory code (Fig. 3f). Interestingly, these selective neurons only made up a small proportion of the dmPFC pseudopopulation (8-9%, Fig. 3f). Using the same cross-temporal SVM analysis as above, we were able to infer that a stable WM code was indeed present. Off-diagonal sample identity decoding accuracy in dmPFC remained significant and strong ( ≥ 80%) from about 700 ms after the delay poke through the end of the delay holding period, suggesting that the small subset of neurons mentioned above was persistently selective (Fig. 3g, center). Conversely, models trained and tested in the MOs and vmPFC decoded sample identity at chance levels over the same duration. To determine the extent to which this persistent decodability relied on the small subset of delay-selective neurons in dmPFC, we searched for neurons from Fig. 3f which were significantly selective for the sample location in at least 20 bins (>= two total seconds) across the entire five second delay period. We found 29 neurons (8.2% of 354) that met this threshold for persistence. Removing these neurons and running the same shuffled cross-temporal SVM analysis from Fig. 3g produced substantially fewer off-diagonal time points with significant decoding of the sample identity (Supplementary Fig. 4), providing more evidence that these persistent neurons are part of a small but significantly stable subpopulation in dmPFC.

Alternative WM mechanisms have also been theorized, which rely on more dynamic representations involving chains of multiple different neurons becoming transiently selective at different points in time, leading to an

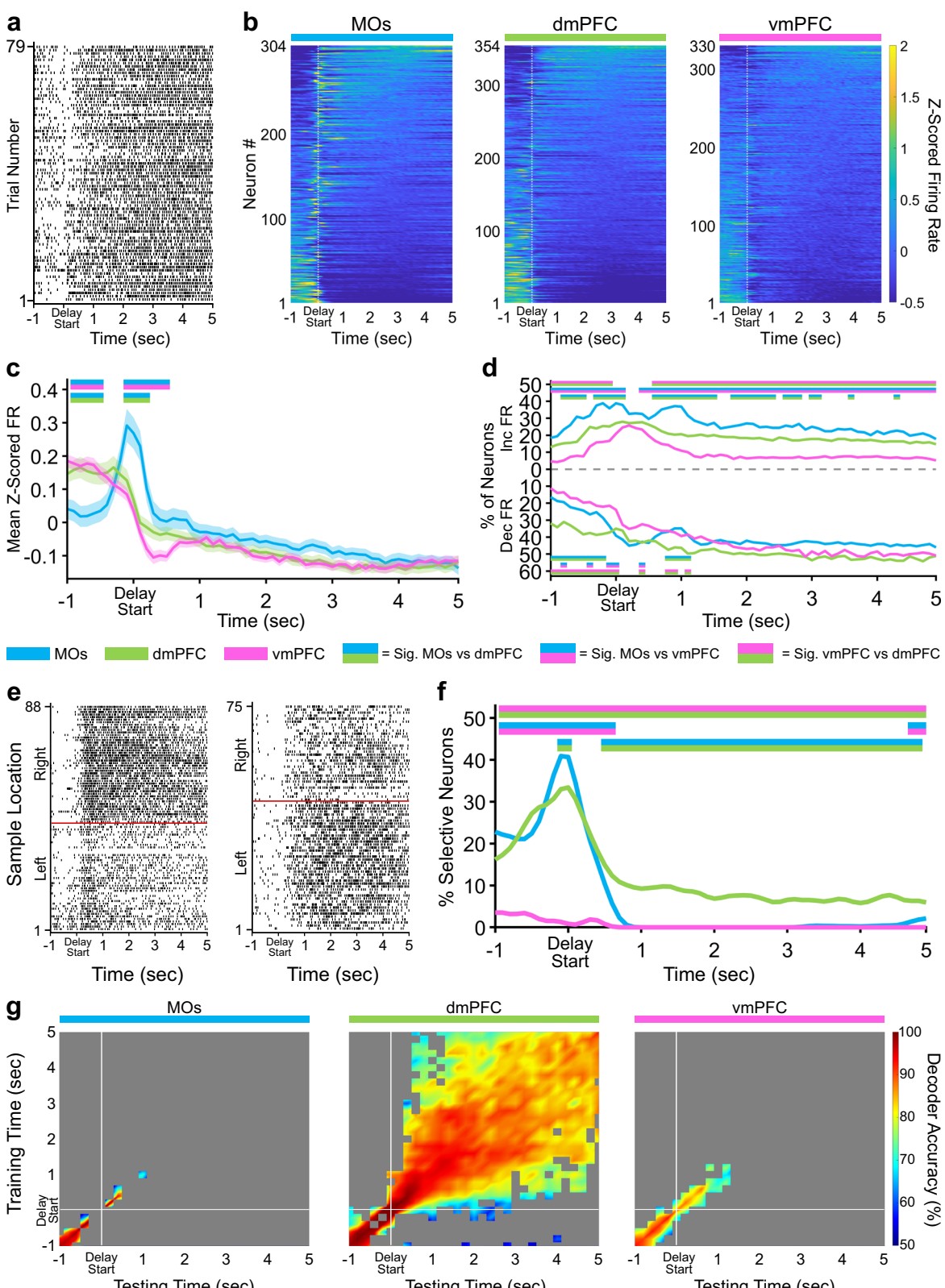

unstable population code[39]. To check if a temporal WM code coexisted with the persistent one, we again removed the 29 persistent neurons from dmPFC and ran a similar delay SVM analysis to the one performed in Fig. 3g. However, instead of training and testing models on all time bin combinations using instantaneous firing rate, we collapsed across time for each neuron and used the bin number with the maximum firing rate on each

trial as the unit of analysis. This transformed the data into a temporal pseudopopulation code for differentiating left versus right sample identity. Essentially, we asked if the times at which the maximum firing rate occurred on all left and right sample trials for each neuron could be used decode the two retrospective locations during the delay. We did this for all subregions and found that there was no significant main effect of subregion on SVM

**Fig. 3 | Delay activity in dorsal mPFC is persistently selective for retrospective sample location. a** Example raster plot from a dmPFC neuron aligned to the start of the delay hold period for all correct trials. **b** Z-scored heat maps of all neurons recorded from each region, sorted by each neuron's mean Z-scored firing rate during the delay phase (5 s period after the white dashed line). **c** The mean Z-scored pseudopopulation firing rate is higher in MOs around the delay poke compared to the other regions. Lighter shaded areas above and below the solid lines represent the standard error of the mean at each time point. **d** The percentage of neurons significantly increasing or decreasing their firing rate from baseline (ITI) is also higher in the MOs at delay start. **e** Example sample location-sorted raster plots of two dmPFC neurons exhibiting persistent selective firing for retrospective right or left sample location throughout the delay phase. Red line separates left from right sample

trials. **f** MOs and dmPFC contain similar percentages of neurons selective for either retrospective sample location (Left and Right combined) in a ~1 s window around the delay poke, but only the dmPFC shows persistent sample location selectivity throughout the entire delay phase. **g** Sample location decoding using a linear support vector machine that was trained and tested on every combination of 200 ms time bins during the delay phase confirms the existence of persistent selectivity only in the dmPFC. White lines indicate the start of the delay period (back port poke). Grey area represents non-significant decoder accuracy compared to a trial identity-shuffled control. In panels **c**, **d**, and **f**, double-colored straight lines represent statistically significant differences (p-value < .05) between the two respective subregions in that time bin, after correcting for both false discovery rate and family-wise error rate.

decoding performance using this temporal procedure (one-way ANOVA, $F_{(2,27)} = 1.29$, $p = 0.29$). The mean, cross-validated, decoding accuracy across 10 subsamples (see Methods) for each region was 51.2% for MOs, 58.5% for dmPFC. and 56.5% for the vmPFC. Thus, a persistent coding scheme, and not a temporal one, seems to be the main WM mechanism used by the dmPFC to maintain retrospective sample identity in this task.

Furthermore, we broke down each pseudopopulation into individual sessions to examine how evenly the persistent dmPFC selectivity was spread across separate recordings (Supplementary Figs. 5-7). This process identified that five out of six dmPFC mice had at least one recording session with sustained sample selectivity. Notably, the single dmPFC mouse with zero stable decoding sessions (Supplementary Fig. 5, dmPFC 4) was the same mouse from Supplementary Fig. 1b which we reasoned, due to its behavior, was likely not using a WM-based strategy to perform the task. This analysis supports our conclusion that the majority of mice perform this DNMTP task using a similar WM-based strategy. Moreover, for each session we plotted the mean cross-temporal SVM decoding performance during the delay against several other session-based experimental parameters (neuron number in each session, individual session task performance, and average choice reaction time). Only the dmPFC contained significant correlations between decoding accuracy and these parameters (Supplementary Fig. 8), strengthening our previous interpretation that dmPFC is the main subregion involved in processing WM representations of sample identity during the delay.

Finally, to substantiate the idea that the mice were actually using this persistent sample identity information during the delay to complete the DNMTP task, we performed a similar SVM analysis to compare the strength of the stable code between correct and incorrect trials. After subsampling to account for the small number of incorrect trials per session, the ability to decode sample port identity was significantly better on correct trials, both in the period directly before the delay poke and during the five second delay itself (Supplementary Fig. 9). Behaviorally, this suggests that incorrect trials might not uniquely arise due to a lapse in WM maintenance during the delay period, but may also be due to a failed encoding of the sample information into WM before the delay even starts.

### Retrospective sample port information is most strongly encoded by the dmPFC during the choice

Reminiscent of nose-pokes in the previous phases, the MOs again contained the largest number of active neurons (Fig. 4a-d). We also recorded considerable activity starting approximately 1.5 seconds before the choice poke which was highest in the vmPFC (Fig. 4c). Around this same time period, the proportion of neurons increasing their firing rate was not different between regions. However, the number of neurons significantly decreasing their firing rate was lower in the vmPFC (Fig. 4d). Contrary to the previous two task phases, retrospective sample location selectivity in this pre-poke period and the period directly around the poke was most represented in the dmPFC instead of the MOs (Fig. 4e, f). This significance continued in dmPFC for about one second after the reward was received, once again implying that the dmPFC most stably represents the location of the sample port that was visited earlier in the trial. Consistent with this, the cross-temporal SVM uncovered more stable significant sample decoding after the

choice poke in dmPFC lasting about two seconds into reward consumption compared to only about 500 ms in the other two subregions (Fig. 4g).

### MOs contains the most explainable firing rate variance around pokes

Although selective coding of retrospective sample port identity is necessary for successful DNMTP performance, mice must also constantly monitor several other task variables to make the correct non-match choice and maximize reward. These include tracking the current phase of the task/location in the box so that the correct motor strategy can be enacted at key time points, anticipating and selecting the upcoming choice, and interpreting the outcome of the choice (correct or incorrect) so that a mouse can update its strategy on the next trial if necessary. Any combination of task variables may interact at any given point during a single trial, so we wanted to analyze how each variable contributed to firing rate variability of individual neurons when taking other variables into consideration. To do this, we needed to use two versions of a general linear model (GLM). The first collapses time by considering all phases of the DNMTP task simultaneously to examine task phase encoding, and the second removes the task phase variable altogether in order to study the remaining variables in more temporal detail over the entire trial.

In the first GLM, we were mainly interested in how transitioning into different task phases (sample, delay, or choice) contributed to the total neural firing rate variability in each pseudopopulation. Because task phases occur sequentially, we needed to create a time-stacked predictor matrix centered around all pokes simultaneously (sample, delay, and choice), so that this variability could be teased apart in the same model (see Methods for details). The matrix contained the following four predictor variables: sample port location (left or right), choice port location (left, center, or right), outcome (correct or incorrect), and poke context (sample, delay, or choice poke). Coefficient of partial determination analysis (CPD, see Methods for details) estimated the proportion of each neuron's total firing rate variability that could be explained by each predictor variable in five separate 200 ms time bins around the time-stacked poke events. When we calculated the total CPD for all variables by summing the means of each variable's CPD across this window, we found that the GLM explained total firing rate variability to the greatest degree in the MOs pseudopopulation, and to the lowest degree in vmPFC (black asterisks, Fig. 5a). Looking at each regressor individually, we found that poke context accounted for the majority of this explainable firing rate variability in all pseudopopulations (colored asterisks, Fig. 5a), indicating that information about task progression and/or the mouse's location in the box is a crucial part of mPFC computation regardless of the subregion. In line with these findings, the MOs had significantly higher CPD values for poke context compared to dmPFC and vmPFC in all time bins around pokes (Fig. 5b).

Next, we assessed how well the population activity in separate subregions could decode the poke context around poke events. We implemented a similar SVM approach as the one described in previous sections, but we applied a multiclass coding scheme instead of a binary one since the model now had three possible choices (sample, delay, or choice phase) to differentiate. Surprisingly, despite the gradient of poke context encoding from MOs to vmPFC described above, the SVM was able to decode poke

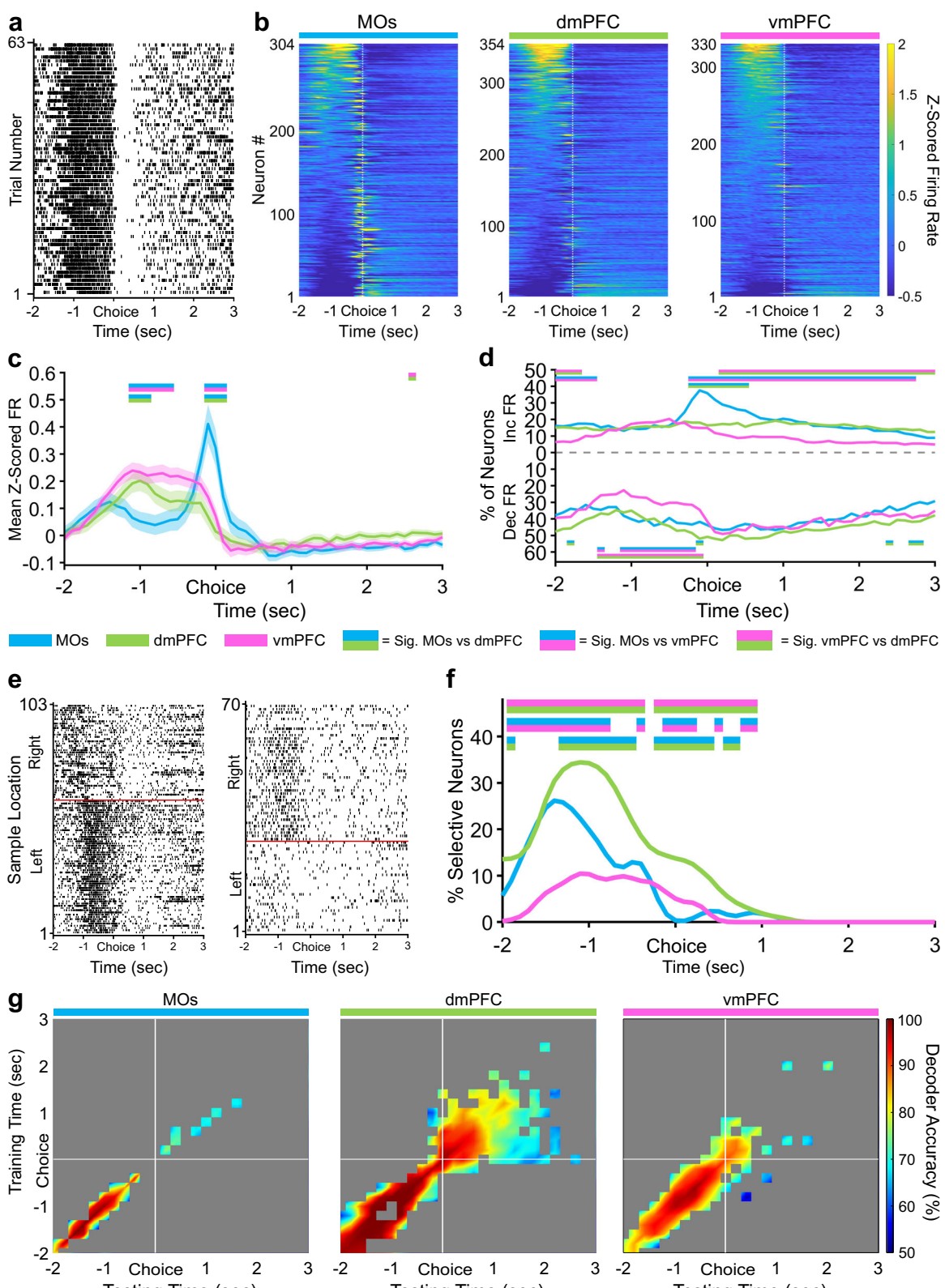

context with 100% accuracy in all subregions (Fig. 5c). To test whether the results in Fig. 5c were due to a saturation effect from a large number of neurons in all pseudopopulations, we ran SVMs on progressively larger subsamples and found that activity in the MOs, which had the highest CPD for poke context, is also better at decoding this context with fewer cells than the other subregions (Fig. 5d).

## A GLM over time uncovers distinct and dynamic subregional processing of different task variables

The first GLM required time-stacking the predictor matrices to assess the extent to which neurons in our pseudopopulations could differentiate between the three task phases. However, with this design, the CPD values for the other three variables may be confounded by the possibility that they are

**Fig. 4 | During the choice phase, vmPFC has a higher pseudopopulation pre-choice Z-scored firing rate, and dmPFC contains more neurons selective for retrospective sample location. a** Example raster plot from a vmPFC neuron aligned to the non-match choice poke for all correct trials. **b** Z-scored heat maps of all neurons recorded from each region, sorted by each neuron's mean Z-scored firing one second before the choice poke (white dashed line). **c** The mean Z-scored pseudopopulation firing rate is higher in vmPFC leading up to the choice compared to the other regions, while activity right around the choice poke is highest in MOs. Lighter shaded areas above and below the solid lines represent the standard error of the mean at each time point. **d** The percentage of neurons significantly increasing their firing rate from baseline (ITI) is higher in MOs at choice poke and the percentage of neurons decreasing their firing rate is lower before the choice in vmPFC. **e** Example sample location-sorted raster plots of two dmPFC neurons exhibiting

transient selective firing for the right or left sample location during the delay phase. Red line separates left from right sample trials. **f** Although activity is higher in vmPFC leading up to the choice poke, dmPFC contains the most neurons with retrospective sample selectivity for either sample location (Left and Right combined) out of all three regions. **g** Sample location decoding accuracy of a linear support vector machine trained and tested on every combination of 200 ms time bins during the non-match choice phase. White lines indicate when the mice poked the choice port and got rewarded. Grey area represents non-significant decoder accuracy compared to a trial identity-shuffled control. In panels **c, d,** and **f**, double-colored straight lines represent statistically significant differences (p-value < .05) between the two respective subregions in that time bin, after correcting for both false discovery rate and family-wise error rate.

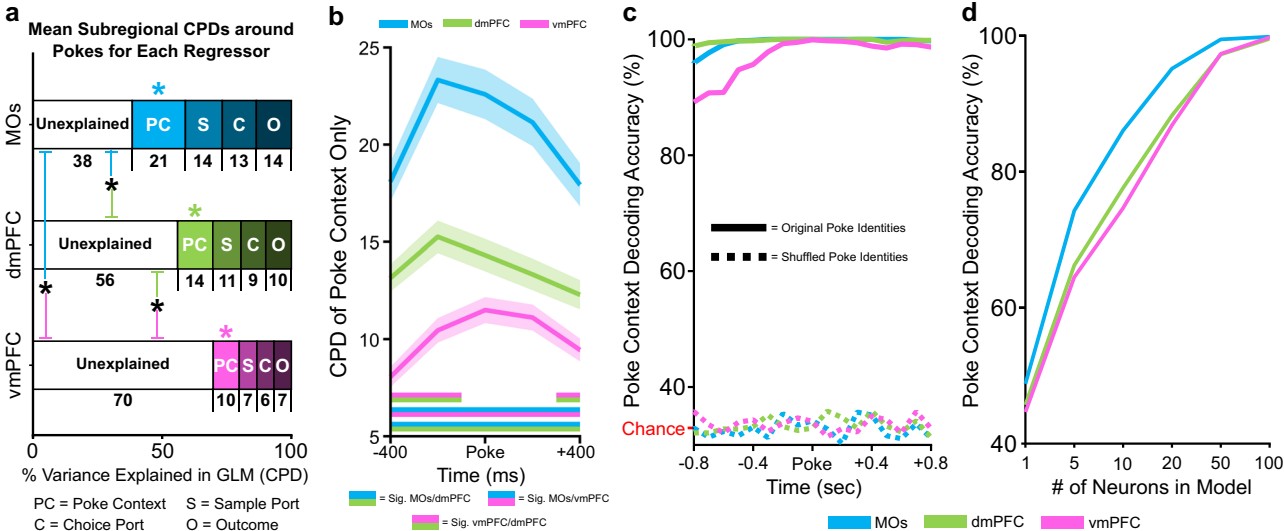

**Fig. 5 | DNMTP task variables account for the most poke-related firing rate variability in MOs. a** The coefficient of partial determination (CPD, calculated for each neuron) from a general linear encoding model was used to estimate the proportion of total pseudopopulation firing rate variance around pokes that was either unexplained (white bars) or that could be explained by our four regressors (Poke Context, Sample Port, Choice Port, and Outcome). Numbers under each regressor represent the mean CPD (% variance explained) across five 200 ms time bins around the poke for that regressor (this time frame can be visualized in the following panel, 5b). A one-way ANOVA revealed a significant main effect of subregion on explainable variance, $F_{(2, 985)} = 44.54$, $p = 3.02e\text{-}19$. Bonferroni post-hoc comparisons found that the MOs had the most explainable variance among subregions, while vmPFC contained the least (black asterisks, MOs vs. dmPFC $p = 3.02e\text{-}7$, MOs vs vmPFC $p = 8.78e\text{-}20$, dmPFC vs vmPFC, $p = 5.35e\text{-}5$). Furthermore, in each subregion, one-way ANOVAs with Bonferroni post-hoc tests found that of the four regressors, Poke Context (PC) was the largest contributor to explained variability.

Colored asterisks represent significant differences within subregions between the CPD values for Poke Context and all three other regressors. **b** Looking at the CPD of Poke Context in isolation at all time points around the poke, we confirm that this regressor accounts for significantly more firing rate variability in the MOs at all time points around pokes, followed by dmPFC and then vmPFC. Double-colored straight lines represent statistically significant differences (p-value < .05) between the two respective subregions in that time bin, after correcting for both false discovery rate and family-wise error rate (see Methods). Lighter shaded areas above and below the solid lines represent the standard error of the mean CPD from the pseudopopulation at each time point. **c** Despite MOs containing the highest CPD for Poke Context, Poke Context is decodable with nearly 100% accuracy in all regions around pokes using a linear support vector machine. Chance level decoding in the shuffled control (dashed lines) is 1/3. **d** Subsampled SVMs revealed that the MOs can decode Poke Context with fewer cells than the other subregions.

best approximated only around a single task phase. For example, neural firing rate variability caused by the outcome regressor likely only makes sense shortly after the choice phase of the task, when the animal either receives a reward or not. Therefore, to investigate the contribution of sample location, choice location, and outcome to firing rate variability with more temporal precision, we needed to remove the poke context variable and unstack the predictor matrices to their original chronological order. This allowed us to run a GLM on all time bins around pokes and observe how the CPD values of these remaining task variables evolved as the mice progressed through the separate task phases (Fig. 6).

Going one step further than the raw CPD analysis performed above, we instead calculated the proportion of neurons in each region that had statistically significant CPD values (according to a shuffled control analysis, see Methods for details) for individual task variables in 200 ms time bins throughout each of the three task phases. This produced results

analogous to our retrospective sample location selectivity analysis using permutation testing in Figs. 2f, 3f, and 4f, which further strengthens these findings (Fig. 6a). The time-based GLM also established that neither upcoming choice port identity nor trial outcome were encoded in the sample or delay phases of the task (Fig. 6b, c). Importantly, we only observed a considerable number of choice-port-selective neurons around the choice poke itself, with the vmPFC displaying the largest percentage (Fig. 6b). Likewise, only after the choice was made were we able to find significant subregional differences in the number of neurons encoding outcome (Fig. 6c). This was a compelling affirmation that the mice were not choosing an incorrect prospective motor plan or specific choice location before they needed to make the actual choice. Surprisingly, the MOs was the region with the most neurons encoding the outcome variable, possibly due to motor activity changing drastically on correct vs incorrect trials as mice were either consuming a water reward by licking

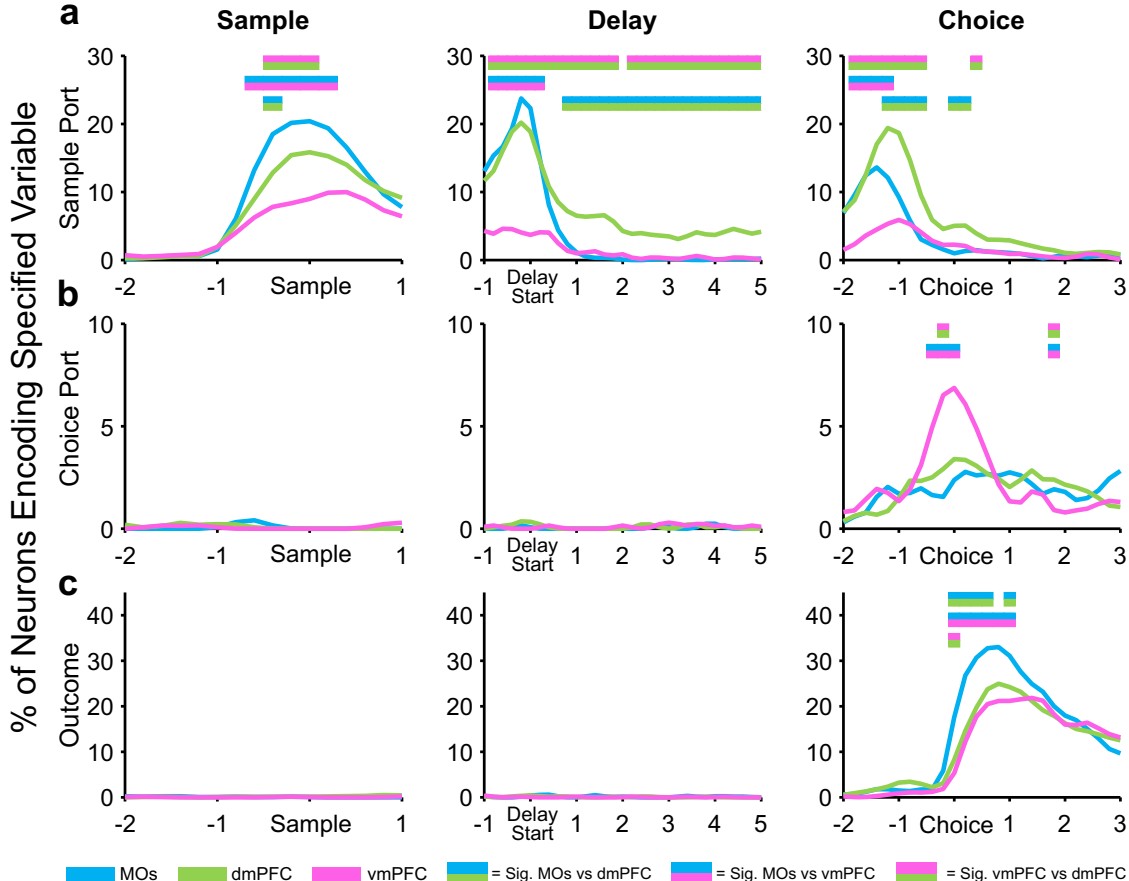

**Fig. 6 | General linear encoding model (GLM) reveals PFC subregion-specific patterns of information encoding across task phases of a working memory task.** After removing the poke context regressor, a GLM uncovers the percent of neurons in each region that significantly encode working memory task variables (sample ports, choice ports, and outcome (correct vs incorrect)) at key timepoints. **a** The region with the highest percentage of significant sample location neurons shifts from MOs to dmPFC as the mice progress through task phases. **b** Prospective choice port location is not detectable in any region until the choice phase, during which vmPFC shows the highest selectivity. **c** The trial outcome (correct vs incorrect), is similarly not encoded by any region until the choice is made. Each region encoded the outcome (correct vs incorrect) after the choice was made, with the MOs having the largest proportion of outcome-encoding neurons. Double-colored straight lines in any panel represent statistically significant differences (p-value < .05) between the two respective subregions in that time bin, after correcting for both false discovery rate and family-wise error rate.

### Tracking GLM beta-weights for retrospective sample location in dmPFC across phases confirms its representational stability

Since sample location was the only task feature significantly encoded by all subregions throughout all phases of the DNMTP task, we wanted to take a closer look at the temporal dynamics of this information in each area. We reasoned that a deeper understanding about how retrospective representations temporally evolve throughout a WM task is critical to fully understand the circuit mechanisms underlying WM, as most definitions of WM rely on the ability of animals to maintain prior information across time. A recurring theme in our data is that neural representations encoding retrospective sample identity seem to be strongly reactivated around subsequent poke events on a given trial. We therefore investigated the extent to which sample identity representations in each subregion were reactivated around these pokes. The percentage of neurons significantly selective for the sample identity around multiple pokes was quantified in the Venn diagrams in Fig. 7a. MOs and dmPFC both had a large number of neurons selective for the sample location across multiple task phases. Interestingly, there were relatively much fewer vmPFC neurons that were sample-selective across multiple pokes (minimal overlap of pink Venn diagram circles).

The maximum GLM beta-weights around all pokes (600 ms before and after poke) from these selective neurons were next identified and converted into vectors. The vectors were sorted by their beta-weight amplitudes for easier visualization, and example histograms of these amplitude vectors from MOs and vmPFC are plotted in Fig. 7b. In these examples, negative beta-weights represented a neuron with higher firing rate on left sample trials, while positive beta-weights signified a higher firing rate on right sample trials. A higher amplitude means the neuron had a higher average difference in firing rate between left and right trials. To compare the subregional stability of these beta-weights across task phases, we ran Pearson correlations between two beta-weight vectors from two subsequent poke events. For example, at the top of Fig. 7b (blue histograms) we correlated the vector of beta-weights from significant sample-selective neurons around sample poke in the MOs (light blue bars) with the vector of beta-weights for those same neurons around the subsequent delay poke (dark blue bars). The $r$ values from these vector correlations are graphed as bars in Fig. 7c for every subregion across every phase-to-subsequent-phase comparison.

This analysis elucidated that the sample-selective neural representations in MOs and dmPFC are similar (i.e., stable) from the sample to the delay phase (example in top panel of Fig. 7b). Conversely, beta-weight vectors from the same two phases in vmPFC are negatively correlated, meaning that neurons in this subpopulation tend to represent the opposite sample location during the delay poke from the one they responded more

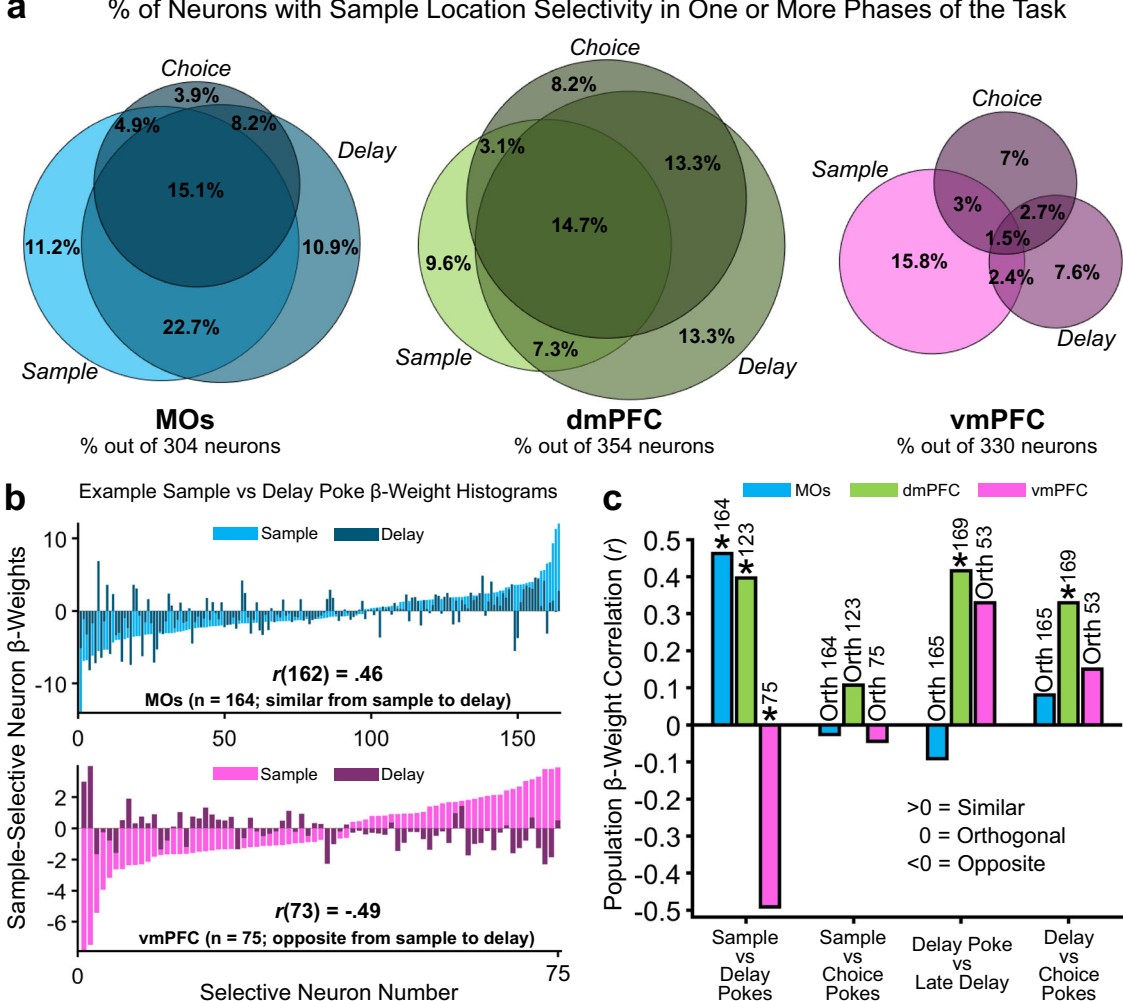

**Fig. 7 | Sample-location-selective firing rate representations have higher stability in MOs and dmPFC compared to vmPFC as mice progress through working memory task phases. a** Venn diagrams depicting the percent of the total pseudo-population in each region that was selective for the sample location around the poke initiating each task phase. Overlapping circles in the Venn diagram represent the percent of neurons that share retrospective sample location selectivity around either two or all three task phase pokes. **b** Example GLM-derived beta-weight histograms from significant sample location-encoding neurons in the GLM showing how most neurons in the MOs remain selective for the same port across sample and delay pokes (top), but neurons in the vmPFC switch initial sample location selectivity from sample to delay (bottom). **c** Pearson beta-weight correlations quantifying the

similarity in location selectivity between pokes initiating one task phase, to pokes initiating another one. MOs location selectivity is similar from sample to delay pokes, but destabilizes over the course of the task. dmPFC remains the most stable across time. vmPFC shifts location selectivity from the sample to delay pokes, but then stabilizes later in the task. Asterisks represent an *r* value that is significantly different from zero, while Orth (Orthogonal) represents an *r* value that is not significantly different from zero. The reason for the non-significant *r* value from the vmPFC correlation in the *Delay Poke vs Late Delay* condition is because of the much smaller number of selective neurons in vmPFC during this time. Numbers above each bar indicate the count of significantly selective neurons in each region used for the correlation.

strongly to during the sample poke (Fig. 7b, bottom). Furthermore, the dmPFC exhibited statistically similar (i.e., stable) sample-selectivity representations across all subsequent poke comparisons, adding more evidence that it uses the most stable retrospective coding of all three subregions (Fig. 7c). Interestingly, none of the areas were stable from the sample to the choice pokes, which potentially arises from the fact that these events are separated in time by a cognitively demanding delay phase, which may largely reorganize neural activity. Overall, the dmPFC seems to utilize short, recurring bursts of selectivity around pokes, with similar selectivity across pokes. This may be evidence of a lesser-studied mechanism of WM called activity-silent WM[40], in which information is stored in the transiently increased synaptic weights of specific patterns of synapses which can be reactivated by a strong input to the WM circuit. However, the data collected in this paper do not allow us to definitively argue that the above results can be explained by this phenomenon.

## vmPFC Z-scored population activity displays the largest change in reward outcome-related firing rate
Up to this point, we had not detected any substantial contributions from the vmPFC to WM performance. However, this region is known to have the densest reciprocal connections with the ventral tegmental area and amygdala out of all PFC subareas[41,42], hinting at potential involvement in more valence or reward-based information processing. As a result, we subtracted the mean Z-scored firing rate of all recorded neurons on correct trials from that on incorrect trials (subsampling for the lower count of incorrect trials), which revealed more pronounced differences in pre-choice and post-outcome population activity in the vmPFC compared to MOs and dmPFC (Fig. 8a, positive values indicate higher FR on incorrect trials). Not surprisingly, the vmPFC was also different relative to MOs and dmPFC in terms of the proportions of neurons responding by either increasing or decreasing their firing rate based on the reward outcome (Fig. 8b).

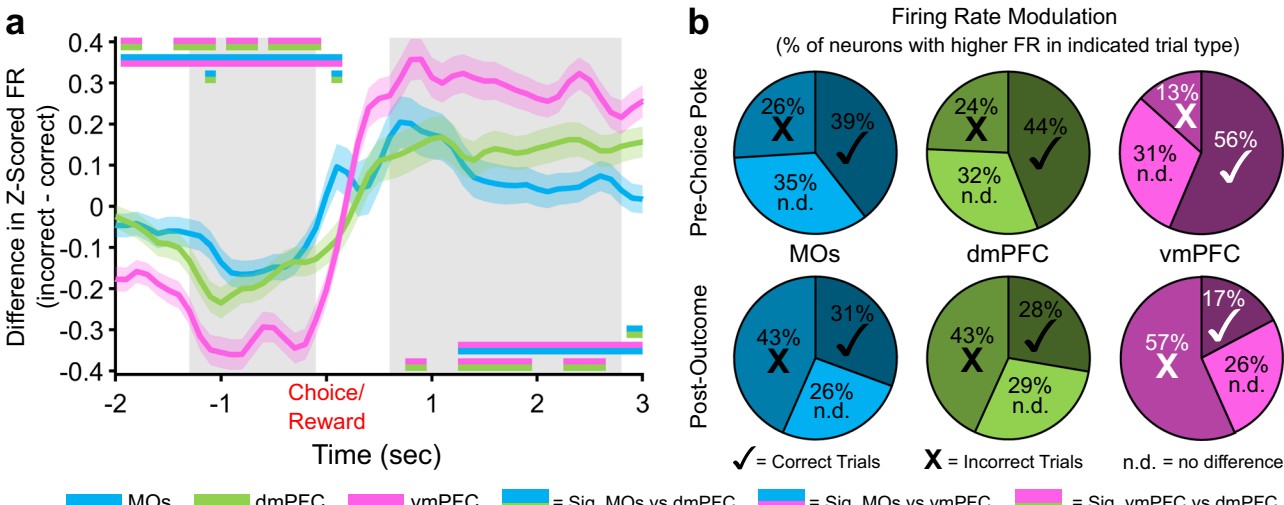

**Fig. 8 | Differences in pseudopopulation activity between correct and incorrect trials are more pronounced in the ventral mPFC before and after the choice poke. a** Z-scored firing rate differences between correct and incorrect trials in each region over time. vmPFC shows the largest decrease in activity on incorrect trials before the poke, and the largest increase in activity in response to an incorrect outcome after the choice. Grey bars represent the time windows used for the pre-choice and post-outcome calculations in panel b. Double-colored straight lines represent statistically significant differences (p-value < .05) between the two respective subregions in that time bin, after correcting for both false discovery rate and family-wise error rate. Lighter shaded areas above and below the solid lines represent the standard error of the mean at each time point. The Choice time point is colored red to signify that this was also the exact time Reward was either delivered (correct trials) or omitted (incorrect trials). **b** Proportions of neurons in each region that have a higher Z-scored firing rate on Correct (✓) or Incorrect (X) trials, or showed no difference in Z-scored firing rate between correct and incorrect trials (n.d.). The proportion of neurons modulating their firing rate on correct vs incorrect trials is also greater in vmPFC. White marks instead of black represent statistically significant proportions of the neurons with higher firing rate on that trial type in the vmPFC compared to the other 2 regions. There were no differences between the MOs and dmPFC.

## Discussion

In this paper, we delineated how the neural populations in adjacent mouse mPFC subregions processed information about WM task-related variables over time. The first key finding was that each subregion exhibited characteristic and differentiated activity during the task relative to the other subregions. The second key finding was that much of the WM task-related activity appeared unrelated to storing WM information (i.e., representations of the previously visited sample port locations). Rather, it reflected changes in activity related to poke context (sample, delay, or choice poke), and rewards. The third key finding was that WM of the sample location could be transiently stored (for less than one second) in both the MOs and dmPFC and in a sustained manner only in the dmPFC, with minimal retrospective representations detected in vmPFC. The final key finding was that vmPFC predominantly represented choice- and outcome-related activity relative to MOs and dmPFC.

Of note, there is a dearth of comparative analyses using single neuron and population activity of neighboring mPFC subregions during WM, particularly in rodents. Activity across mPFC subregions has been examined in rats performing DNMTP tasks[43,44] and a more complex match to place task[45]. In these studies, single neurons responded prominently to reward locations and task phase, but in contrast to the findings presented here, no sustained neural activity related to retrospective sample port location was noted during the delay period. Expanding on these studies, our work implements complementary computational analyses to further explore the disparate functional roles and temporal dynamics/stability of subregional mPFC neural populations during WM. In addition, we included another brain region in our experiments, the MOs, after recent evidence that it supervises more abstract functions than basic motor control and should be considered a part of the mPFC[46].

In our DNMTP task, the MOs behaves in a manner consistent with the theory proposed by Barthas and Kwan[46] which posits its involvement in context-dependent selection of motor plans[47,48] and the online monitoring of sequential sensorimotor tasks[49]. Activity in our recorded MOs neurons peaks in a tight window around all port pokes. Importantly, population-level differences in this transient activity can simultaneously differentiate which part of the task the poke is occurring in alongside distinguishing the left versus right sample port location. These transient poke-centered phenomena may represent close monitoring of the phase of the DNMTP task the animal is in at fixed intervals, while also providing information at each stage about the spatial rule of the current trial, facilitating ongoing motor plan selection and timing. Consistent with the notion that MOs is involved in context-dependent selection of motor plans, it has been reported that MOs can relay its contextual/motor information to many areas of the neocortex it connects with, including sensory and primary motor cortex[50]. Future timed inactivation studies may be useful in exploring these ideas further in WM tasks.

With respect to retrospective sample location selectivity, our analyses uncovered strong sample identity representations in both MOs and dmPFC during all task periods. The retrospective sample location was detectable using single neuron and population analyses, including using an SVM to decode the sample port location and developing a GLM to show that this port location information is significantly encoded in MOs and dmPFC neurons well after the sample phase. As mentioned above, the retrospective sample location-related activity in MOs manifests only briefly around pokes (< 1 s), and these recurring transient patterns of selectivity do not remain stable throughout a trial. We speculate that MOs may not contain the molecular or circuit architecture necessary to maintain persistent activity in a group of neurons[51]. In contrast, subsets of dmPFC neurons are selective for the retrospective sample port at the sample, delay and choice pokes in similar patterns, with a smaller group (8–9%) exhibiting sustained retrospective sample port selectivity throughout the delay. These findings are consistent with the notion that there are overlapping ensembles of WM neurons operating with different dynamics in different PFC subregions on different timescales. They also suggest that briefly active WM ensembles may be more common than sustained ensembles. It will be useful to explore these ideas further in primates and rodents using different WM tasks with a range of delay lengths.

Perhaps the most intriguing finding in our data is the aforementioned stability of retrospective sample identity representations in dmPFC throughout the course of behavioral trials, especially during the majority of

the delay holding period when neither of the other subregions display sample location-related activity. A substantial body of evidence has set the precedent for the existence of persistently selective delay representations in various primate brain areas including dorsolateral prefrontal cortex[52,53] and in head-fixed mice. The latter work was done in the nearby anterior lateral motor cortex (ALM) during a delayed motor response task showing sample-related preparatory activity for an upcoming left or right lick[30–32,54,55]. This task differs from ours in that the head-fixed animals know the exact location of the correct choice throughout the delay period, allowing them to make a precise motor plan which may be represented as persistent activity. In our task, the prospective location that will be rewarded cannot be anticipated. Additionally, the delay lengths used in the reports cited above are relatively short (1–2 s) compared to ours at five seconds. We believe that this longer interval is crucial to unraveling the dynamics of WM, as we observed that most neurons coding for the retrospective sample location in MOs and dmPFC around the delay poke are not sustained beyond 1–2 s into the delay, with a small group of neurons ( ~8%) in dmPFC exhibiting sustained WM activity. Other studies in freely-moving rats also found non-sustained sample-selective delay activity in the frontal orienting field during a WM motor planning task[47], and in prelimbic cortex in a delayed alternation task[56].

An often-overlooked aspect of WM is the need to maintain some signal which updates strategies based on feedback from reward outcomes, so that behavior can be adjusted in the immediate future if results don't match expectations. Our findings in vmPFC align with the possibility that this region is predominantly involved in processing choices and outcomes. This could be communicated by different release patterns of dopamine in this region[57], or changes in firing from glutamatergic amygdala inputs[58]. Although we did not record from all subregions simultaneously, our results allude to a dorsolateral to medioventral trajectory of information processing from MOs to vmPFC as the mice progress through the task. Similar dynamics across distant brain regions have been characterized before in humans[22], monkeys[59], and mice[30], but they remain poorly characterized both in primates and mice in PFC subregions involved in WM, providing an important future direction of study.

One caveat of this study is that these subregions were not recorded simultaneously due to the technical difficulty of probing multiple sites along the curvature of the cortex. Future studies would benefit from using more advanced electrophysiological setups with multiple shanks (e.g., Neuropixels 2.0 probes[60]) to record all subregions at once. We also did not collect video data to accompany these recordings. In future studies, video would be useful for connecting specific movements to associated neural activity patterns[61]. Furthermore, with the way our task was designed, we cannot definitively determine if the mice are actively remembering the retrospective port location or prospectively planning a left or right turn *away* from the sample port during the delay period[16,62], although we argue that both of these scenarios require a WM mechanism. However, our GLM data showing no neurons in any region that were selective for the upcoming choice port lends support to the idea that the sustained representation we describe is most likely a retrospective one that aligns well with stored WM information very closely related to the sample port visited earlier in the trial.

One ongoing challenge in neuroscience is the mapping of rodent mPFC subregions to the corresponding areas from primate PFC. Although this can be done according to common afferent and efferent projections, molecular expression, and functions[63–66], there is often a lack of consensus regarding the extent of similarity/homology. Another challenge has been the difficulty in localizing persistent neural activity in the PFC of freely moving mice performing WM tasks in order to study the underlying mechanisms. By identifying such activity, as well as other types of WM task-related activity, this study provides a foundation to perform such studies using tools that are uniquely available in mice. Our work here also paves the way for more detailed analyses of these networks and a more nuanced and dynamic view of how neighboring brain regions process information in complex and complementary ways as mice progress through a complicated behavior that requires WM and other cognitive functions.

## Methods

### Animals

All animal experiments performed in this study were approved by the Veterans Affairs Portland Health Care System Institutional Animal Care and Use Committee. Nine female and seven male mice, bred on a C57BL/6 J background, were housed in the Veterans Affairs Portland Health Care System Veterinary Medical Unit on a reverse 12-hour light cycle with lights turning off at 8:00 AM (PST), and on at 8:00 PM (PST). All mice were group-housed before electrode implantation, after which they were single-housed until the completion of the experiment to prevent them from damaging each other's implants. Mice were 3 to 4 months old at the start of training, and testing completed at ages ranging from 6 to 9 months. Mice had *ad libitum* access to water (unless restricted for experimentation) and PMI PicoLab 5L0D Laboratory Rodent Diet (LabDiet, Inc., St. Louis, MO, USA). Mice were housed in rooms with constant temperature (22–26 °C dry bulb) and humidity (30–70%) monitoring. We have complied with all relevant ethical regulations for animal use.

One day prior to the start of initial behavioral training, mice were water restricted to 85-90% of their initial body weights so that they were motivated to seek out water rewards. This weight-based water restriction continued for the duration of active behavioral experimentation (but not during recovery from surgery), after which they were promptly returned to ad libitum water until they were sacrificed for implant location confirmation. The long-term water restriction protocol consisted of giving the mice about one gram of a 98%-water, gelatinous hydrocolloid mixture (HydroGel®, ClearH20, Westbrook, ME, USA) daily, after behavioral tasks, to keep the mice at a constant water-motivated weight. Furthermore, during the one-week recovery from surgery, mice were supplemented with an electrolyte-based recovery diet (DietGel® Recovery, ClearH20, Westbrook, ME, USA).

### Behavioral setup and training

On the first day of delayed-non-match-to-position (DNMTP) behavioral training, water-restricted mice were acclimated to the behavioral chamber (Fig. 1a, Bpod, Sanworks LLC, Rochester, NY, USA). The main hardware components of this chamber included a state machine control board and illuminable, photo-gated, water-dispensing ports. Integration of this system with MATLAB software (MathWorks, Natick, MA, USA) and our electrophysiological recording system allowed for precise closed-loop control over specific DNMTP task parameters via custom MATLAB scripts. These parameters included the timing of task phases, lighting of ports, delivery of water rewards or signaling of incorrect behavior with negative reinforcers, and online synchronization with electrophysiological data for accurate timestamping of neural firing and important behavioral events.

After a 15-minute habituation session on day one, mice were taught on day two that only lit ports could dispense water rewards. To do this, we randomly lit and pre-baited one of the four (three "front" and one "back") ports so that water was available as soon as the port light turned on. This allowed mice to initially learn the simple Pavlovian association that water was available from lit, but not dark, ports. Once the mice were familiar with this association (usually after one 15-minute session), we changed to a slightly more instrumental design where water was not dispensed until the mice poked their nose in the lit port and broke the plane of the infrared photogate, prompting them to learn that their active engagement with the port was required for water to be dispensed.

The next step in training was a modified version of the final DNMTP task with intertrial intervals (ITIs), sample phase, delay phase, and choice phase. After a five second ITI, either the left or right front port had a 50% chance to randomly light up on a given trial (sample phase, the center port was never used as a sample location), and the mouse was required to poke its nose in the lit port to get a small water reward (3 μL) and activate the back delay port. The mouse then turned around to poke in the back delay port, which dispensed a one μL reward immediately after the poke. Poking in any of the dark ports during the sample or delay phases led to a punishment consisting of illumination of a bright house light and a behavioral timeout for 15 s, after which the exact same trial restarted. Successful progression

through the delay led to the choice phase, where the previous sample port lit up along with one of the other two front ports (randomly, 50% chance). The mouse had to poke the lit port that it did not previously visit during the sample phase to complete a correct non-match choice and get a larger 7 μL reward. Consumption of this reward typically took about two to three seconds, after which the mice left the port. Leaving the port after water consumption initiated a five second ITI period in which the mouse could not enter any other ports, or the ITI timer would reset. An incorrect choice similarly led to an illumination of a bright house light and a fifteen second timeout, but in this case a new trial with new port locations started afterwards.

Once mice achieved 70% non-match performance on this training task with no delay, we removed the sample phase reward and implemented a gradual increase of delay length in each session to help the mice get to the final delay period of five seconds. The mice had to hold their nose in the back port until the delay timer ended to get the small 1 μL reward and enter the choice phase. After each successful delay hold, the timer went up from zero by 0.10 s until a five second delay was reached. We let the mice do this until they reliably got to 5 s for the delay period. The final version of the task had the mice starting the delay at zero seconds and walking up by one second per trial to the final five second delay. These first five trials were removed from analysis. The final task sessions, in which we also recorded brain activity, usually lasted around one hour and the mice completed anywhere from 33 to 168 trials (mean of 106.43 trials with a standard deviation of 27.35 trials), depending on motivation for water based on hydration status. Once mice performed at >70% for three consecutive sessions they were implanted with electrodes. All behavioral analyses were from sessions where neural recordings also took place.

## Electrode implantations and single-unit recordings
Custom 28-channel implantable electrode bundles were constructed in-lab. The process consisted of threading 32-channel electrode interface boards (EIB-36-Narrow-PTB, Neuralynx, Inc., Bozeman, MT, USA) with 12 μm diameter tungsten wire (California Fine Wire Company, Grover Beach, CA, USA), and securing the wires in the board with gold pins (Neuralynx, Inc., Bozeman, MT, USA). Four slightly larger diameter local field potential wires were implanted in various mPFC-connected brain regions, although none of the data collected with these wires was used for analysis. Silver ground and reference wires were also soldered onto the EIB. The apparatus was built on a custom 3D-printed scaffold (Grey V4 resin, Formlabs, Somerville, MA, USA), with holes for drivable screws (McMaster Carr, Elmhurst, IL, USA) that allowed for advancement of the electrodes after every recording session. The wires were affixed to the EIBs using dental cement (UNIFAST Trad, GC America Inc., Alsip, IL, USA) to protect them from damage.

For implantation, mice were lightly anesthetized with 3% vaporized isoflurane (Covetrus, Dublin, OH, USA) and transferred to a stereotaxic surgery apparatus (David Kopf Instruments, Tujunga, CA, USA) where they were kept at ~1% isoflurane for the remainder of the surgery. Body temperature was monitored with a Physitemp (Clifton, NJ) TCAT-2LV temperature controller system, which held animals between 36 and 37 °C. Prior to initial incision, they were injected with carprofen and dexamethasone for pain management, along with a topical application of lidocaine to the skull and surrounding skin. Electrode bundles were implanted on the left side of the skull at the following coordinates (from bregma): supplementary motor area (MOs): +1.80 mm anterior, -1.50 mm lateral left, -1.00 mm ventral to brain surface; dorsomedial prefrontal cortex (dmPFC): +1.80 mm anterior, -0.40 mm lateral left, -0.50 mm ventral to brain surface; ventromedial prefrontal cortex (vmPFC): + 1.80 mm anterior, -0.40 mm lateral left, -1.70 mm ventral to brain surface. A larger diameter reference wire was implanted in the left striatum: + 0.50 mm anterior, -1.60 mm lateral left, -2.50 mm ventral to brain surface. This reference location was chosen due to its large size, making it an easily reproducible target to reliably hit during surgery. A ground screw was also placed in the skull over the right (contralateral) cerebellum. The full setup was secured to the skull using the same dental cement mentioned previously. This included threading screws into

skull-secured acrylic cuffs so that they could be advanced and drive the electrodes deeper into the brain.

Mice were allowed to recover for at least one week with daily health monitoring before returning to water restriction and behavioral testing. Electrical recordings during behavior began after two to three re-habituation sessions while plugged into the electrophysiological tether and commutator (Doric, Quebec, Canada). Data was collected with a CerePlex Direct neural acquisition system connected via a 32-channel CerePlex μ (mu) headstage (Blackrock Neurotech, Salt Lake City, UT, USA) to the implanted EIB. Unfiltered data was sampled at 30 kHz throughout an entire behavioral session. Single-units were isolated offline using Kilosort3 and timestamped to behavioral events. Since the geometry of recording sites in our bundle was unknown, we used a random linear arrangement for our probe configuration parameter and turned off the registration and drift-correction parameters. Units considered "good" by the Kilosort algorithm (consistent unique waveform shape and clean autocorrelations) were then manually curated and removed if their amplitudes or template shape drifted significantly over the course of the recording, or if manual offline cross-correlations with other units determined that they were duplicates. In the second case, the highest amplitude duplicate was kept, and the rest were not used in further analyses. Similarly, any isolated units with a mean firing rate of < 0.5 Hz across the entire session were also excluded from further analysis.

## Statistics and reproducibility
The following analyses were completed using custom MATLAB (R2022a and R2024a) scripts, and the use of specific built-in MATLAB functions is noted when appropriate. These analyses were performed with the goal of testing for differences between these subregional pseudopopulations.

## Z-scoring and comparison of Z-scored firing rate across subregions
After spike sorting, we created three pseudopopulations by combining all neurons within each of the PFC subregions across all recording sessions. The total neuron count for each pseudopopulation was 304 from the MOs, 354 from the dmPFC, and 330 from the vmPFC. The first analysis involved Z-scoring the firing rate of every neuron around key behavioral events. This was done by summing the spikes in every 100 ms time bin five seconds before and after the sample, delay, or choice pokes for every correct trial (one hundred total bins for each poke). We then normalized each trial of each neuron across the time bins to create time-based Z-scores. These Z-scores were averaged across all correct trials in that neuron's session to get the mean Z-score for every neuron around important DNMTP task events, and this result is depicted in the heatmaps seen in Figs. 2b, 3b and 4b. We could then take the mean across all neurons in each pseudopopulation to see how the regions differed in terms of simple firing rate changes over time (Figs. 2c, 3c and 4c).

To evaluate if the pseudopopulation Z-scored firing rate differed between subregions, we ran a separate one-way ANOVA (MATLAB function *anovan*) at every relevant time bin shown in the figures (this number was different between task phases). After uncorrected p-values for each ANOVA were found, we adjusted them for false discovery rate (FDR) using the Benjamini-Hochberg method[67]. Each time point that still had a corrected p-value of < .05 was taken, and unpaired t-tests in that bin were conducted on each combination of subregional comparisons. The p-values from these multiple comparisons were then Bonferroni post-hoc corrected, and only comparisons with adjusted p-values still below .05 were considered significantly different at the corresponding timepoint. The above ANOVA strategy was also used to calculate significant subregional differences in the changes in Z-scored firing rate on incorrect vs correct trials in Fig. 8a, except that that correct Z-scores were subtracted from incorrect Z-scores before analysis. In these time binned ANOVAs, the within-subject degrees of freedom is 985 (total number of neurons across pseudopopulations minus 3 groups), and the between-subject degrees of freedom is 2 (number of pseudopopulations minus 1).

We took a related approach to analyze subregional differences in the proportions of neurons exhibiting a significant increase or decrease in

firing rate. We first used a Wilcoxon signed-rank test to compare Z-scored firing rate in a one second baseline period during the intertrial interval, to the Z-scored firing rate at each 100 ms time point around important task events (Figs. 2d, 3d and 4d). Then, instead of running ANOVAs to check for differences between groups at each time point, we ran a $\chi^2$ test for homogeneity of proportions across the three groups (MATLAB function *crosstab*). Like the ANOVA approach, we also adjusted p-values for false discovery rate over time, again using the Benjamini-Hochberg method. If adjusted p-value was still less than .05 for a time bin, we ran separate pairwise $\chi^2$ tests for each combination of subregions and corrected for these multiple comparisons using the Bonferroni-Holm method[68]. Any adjusted p-value below .05 after these conservative corrections was considered to represent a significant difference in the proportion of neurons either increasing or decreasing their firing rate from baseline (during the intertrial interval) between two groups at that time point. Moreover, this $\chi^2$ strategy was similarly employed in Fig. 8b, to study the differences in the proportions of neurons that increase or decrease their firing rates in response to incorrect trials compared to correct ones.

### Determining retrospective sample location selectivity using permutation testing

Retrospective sample location selectivity was analyzed in several ways throughout the paper. In Figs. 2f, 3f and 4f, we used a permutation testing method to compare the raw spike counts in 100 ms time bins between left and right sample location trials (correct trials only). Since sessions rarely had an equal number of correct left and right sample trials, for each session we randomly subsampled trials from the greater of the two to match the number of trials from the lesser. Next, we randomly sampled two trials from a combined subpopulation of left and right samples, such that the spike counts for these random two trials could be from two left trials, two right trials, or a left and a right trial. We took the difference between two randomly sampled trials 1000 times and created a shuffled distribution of differences. We then calculated the true mean difference between all left and right trials, compared that to the shuffled distribution, and counted the number of shuffled differences that were greater than or less than the true mean difference. Neurons were considered to be significantly selective for a sample location if < 25 out of 1000 shuffled differences were greater in magnitude than the actual difference (approximating a two-tailed p-value of < .05). This was done for every time point around key poke events. Significance between subregions was determined using a similar $\chi^2$ strategy to the one mentioned in the above section.

### Support vector machines (SVMs) for cross-temporal decoding of sample location and poke context

To track the stability of location selectivity across time in each region, we trained linear SVMs (MATLAB *fitcsvm*) to decode retrospective sample location every 200 ms using binned raw spike counts from each neuron in a given region's pseudopopulation on all correct trials. We then assessed how well each time point's trained model could predict sample location based on the pseudopopulation activity at all time points (including the one it was trained on). A stable population representation would display above-chance predictive accuracy ( > 50%) at time points far from the training time. A leave-two-out cross-validation scheme was applied to protect against overfitting the model. This consisted of holding out one trial from both left and right location samples and testing how the model trained on the remaining trials predicted the identity of these held out trials. Importantly, we subsampled each session's left and right trial counts to 13, which was the lowest left or right sample location trial count across all sessions, although most sessions had many more than 13. Our leave-two-out strategy was therefore repeated 13 times, with each value from this subsampling appearing once without replacement in the testing set. The prediction accuracy of the model for each subsample was calculated as the number of correct classifications of all held out trial combinations, out of 26. The overall subsampling procedure was repeated 10 times, for a total of 260 model predictions to calculate non-shuffled prediction accuracy for each cross-temporal comparison. For each of these 10 subsampling repetitions,

we also randomly shuffled the trial identity and ran the above procedure 100 more times to generate a false distribution of predictions to compare the non-shuffled accuracy value to. This led to a total of 1000 shuffled to non-shuffled comparisons. If the mean (out of 260 predictions) prediction value from the non-shuffled "true" model was greater than at least 950 of the shuffled model predictions (approximating a one-tailed p-value less than .05), this cross-temporal time point was considered significant. Any non-significant time point was represented in the figures as a grey square covering that cross-temporal location on the graph (Figs. 2g, 3g, and 4g).

A second pseudopopulation SVM was used in Fig. 5 to classify the poke context (sample, delay, or choice poke) in a brief window around the three poke events across correct trials. Because there were three contexts, a multiclass SVM was needed. We used the *fitcecoc* MATLAB function with a 'one-vs-one' coding design and 5-fold cross validation per comparison, which produced a chance level of 1/3. We also ran a shuffled version of this model to confirm that the model was not overfit. The reason these SVMs were only done in a short time window around the pokes was to maximize confidence that the phase we were testing did not contain any residual information from a prior or future task phase, since we could not be sure of the exact time an animal was aware of its transition to another poke/task phase.

### General linear modeling

We used two separate general linear model designs (GLM, MATLAB function *fitlm*) to characterize how neurons in each region encoded multiple DNMTP task variables simultaneously. The first GLM analysis was mainly designed to determine the extent to which neurons in each pseudopopulation encoded the task phase that mice were in around pokes (sample, delay, or choice, Fig. 5). This was done in a brief time window centered around poke events (-400 to +400 ms) since mice took inconsistent and unpredictable amounts of time to progress to different phases, and we wanted to make sure the task phases in question were temporally well isolated. For this GLM, the predictor variables included poke context (sample, delay, or choice poke), sample location identity (right or left), choice location identity (left, center, or right), and outcome (correct or incorrect). Construction of the GLM predictor matrix was done by converting the levels of these regressors (for example, left or right for sample location) into dummy variables for each neuron (MATLAB function *dummyvar*, in the above example, left sample trials become 1 and right sample trials become 0 for an individual session). When training a GLM model to predict neural firing rate, each of these one-hot encoders was treated as a categorical predictor variable. In the first GLM analysis, the three predictor matrices for task phase were stacked on top of each other such that poke time was collapsed. Since we were doing this for individual neurons, the length of each stacked predictor matrix varied, and was equal to the number of trials in a given neuron's session multiplied by 3 (for the 3 stacked task phase matrices).

For model training, we took a similar overall GLM approach to Akam et al. [69], in which a new model was trained to on every time bin to predict the firing rate of each neuron based the levels of all predictor variables at that time point (known as the "full" model). The sum of squares error (SSE) of the full model represents the amount of residual variance in a neuron's firing rate that cannot be explained by changes in the level of predictor variables. The full model should account for the most firing rate variability in a given neuron and should therefore have a relatively low SSE since it contains the most sources of potential variability. To uncover the extent to which pseudopopulations encoded singular predictor variables, we found the coefficient of partial determination (CPD) for each individual variable. This can be calculated by removing the singular predictor variables from the full model and running another GLM on the reduced model. Since the reduced model should have a larger error term than the full model, subtracting the full model from each reduced model should approximate the contribution of each removed variable to the full model's firing rate prediction. The resulting number is the CPD for that singular variable, and it approximates the percentage of total firing rate variability for a given neuron explained by that variable. A CPD was generated for each neuron at each time point, and

the CPD for each pseudopopulation was averaged across neurons and reported over time (Fig. 5b) or taken as a mean across the time points immediately surrounding the poke (Fig. 5a). Using CPDs from all predictor variables, we could then compare the contributions of each variable to the explainable firing rate variance in each region.

For our second GLM analysis (Fig. 6), we were instead interested in whether each pseudopopulation dynamically encoded the other three non-task-phase variables across the different task phases. For this, we had to remove the task phase regressor from our stacked predictor matrix and subsequently unstack it so that this analysis could be done on each time point around all task phases and likewise compared across task phases. Thus, each predictor matrix now only contained three predictor variables and was 1/3 the size of the stacked matrix for each session. Statistically significant CPDs for each predictor variable at every time point were determined by comparing the true CPD value to a shuffled distribution of CPDs generated from randomly shuffling predictor variable identities 1000 times. As a very conservative cutoff, CPDs were only considered significant if none of the shuffled CPD values were higher than the unshuffled one, and we calculated the proportion of neurons at every time point that fulfilled this criterion. We then could determine differences in the proportions of neurons exhibiting complex DNMTP task variable encoding between regions using the same $\chi^2$ approach employed at the end of the section *Z-scoring and comparison of Z-scored firing rate across subregions* and follow these proportions over time around the poke events of all task phases.

Among the most informative GLM outputs are beta-weights representing the strength and direction of the relationship between the firing rate of every neuron and the levels of each predictor variable. As another approach to visualize the stability of retrospective sample port location encoding, we recovered the maximum beta-weight amplitudes for every neuron with significant selectivity in a 600 ms time window around pokes. In this case, the beta-weights of significant neurons could be positive or negative with respect to the sample port location predictor variable, with negative signifying neurons that fired more on the left side of the box and vice versa. We quantified how these beta-weights changed over time using Pearson correlations (*r*, MATLAB function *corr*) comparing the significant beta-weight population vector at one time point to the beta-weight population of those same neurons at a future time point. The significance levels of these correlations were also reported in Fig. 7c and represent the confidence that the reported correlations are different from zero.

## Reporting summary

Further information on research design is available in the Nature Portfolio Reporting Summary linked to this article.

## Data availability

The source neural and behavioral data used for all analyses in this study are publicly available in a Figshare repository with the identifier https://doi.org/10.6084/m9.figshare.26999575.

## Code availability

Custom MATLAB functions for the SVM and GLM analyses in this study are publicly available in a Figshare repository with the identifier https://doi.org/10.6084/m9.figshare.26999575. All other custom code used in this study will be made available by the corresponding author upon reasonable request.

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

## Acknowledgements

We would like to thank Bakr Alkarawi for his help assembling the electrode implants used in this study. We also thank Michelle Palumbo for assistance with BioRender. A.I.A. and L.B. were supported by the Portland VA Research Foundation, the OHSU Physician-Scientist Award, and IAAs D-22-OD-0001, ADA12013, CDER-20-I-0546. A.S. and R.M. were supported by NIAAA T32 AA007468. R.J.O. was supported by NIDA T32DA007262.

## Author contributions

A.S., A.I.A. and L.B. conceived and designed the experiments; A.S. and L.B. performed the experiments; A.S., A.I.A., R.J.O. and R.M. analyzed the data; A.S. wrote the initial draft of the manuscript; A.S., L.B., R.M. and A.I.A. revised the manuscript and approved the final version before submission.

## Competing interests

The authors declare no competing interests.
