## [Transparent Peer Review File · Communications Biology]

Reviewers' comments:

Reviewer #1 (Remarks to the Author):

The authors present a highly interesting, excellently conducted and analysed study revealing key neural mechanisms of working memory (WM) in mice. They record single-cell ensembles in three frontal regions, the supplementary motor area (MO), the dorsomedial PFC (dmPFC; although according to coordinates that might rather be ACC/Cg1 or Cg1/PrL border) and the ventro-medial PFC (vmPFC, akin to classical IL). Importantly, they do so in an advanced, operant rodent WM-task, that (compared to commonly used, simple alternation-based maze-assays) is more similar to human WM tasks and specifically to delayed-choice (as opposed to delayed-response) tasks, which afford better control over task timing and other psychological variables (attention, motivation) and allow to better discriminate actual memory retention from preliminary motor-programming. In addition to introducing and using this very informative task, the authors also make important revelations regarding distinct roles of the investigated areas; most importantly they show that MO contains the strongest encoding of the stimulus-identity = spatial motor action (poke-hole) during and several seconds after poking, but only the dmPFC retains the to-be-memorized stimulus representation throughout all of the very long delay, functioning as the actual memory store - and seemingly largely based on persistent activity (see the debate <https://www.jneurosci.org/content/38/32/7020> vs. <https://www.jneurosci.org/content/38/32/7013> for reference). During choice, vmPFC (in addition to the other areas) also becomes highly active and encodes the to-be chosen poke-hole, which is also a discrimination between rewarded and non-rewarded choice. Therefore, one could imagine a functional distinction between such areas as MO - as motor area - encoding the spatial determinants of the motor action before, during and after poking-action, dmPFC (or Cg1) encoding the actual memory in absence of motor/sensory information, and vmPFC possibly encoding reward-related aspects.

I have no major reservations against publication of this excellent manuscript in Communications Biology, but have a few minor suggestions:

1. Figure 2: It could be worthwhile having some statistical measure for the the share of neurons modulated by an event in each of the three brain regions, to know if some neurons can be identified as activated or inhibited during the sample-phase poke with some statistical certainty. It is unclear to me, to what extent the analysis in 2D contains such statistical certainty and what the actual comparison interval for the 0.5 Z-unit increase is and what the p-value stated in l. 98 and indicated in 2D actually refers too. E.g. one could

use a within-subject Wilcoxon-test across trials comparing amplitudes in a baseline interval to a certain time window around the poke (<http://www.sciencedirect.com/science/article/pii/S0092867417312047>). Using a Chi-square test comparing the share of activated/inhibited/unchanged neurons one could get a second indication for the statement that MO differs from the two PFC region, but not the latter from each other.

2. Figure 2G. The cross-temporal SVM is a very interesting and revealing approach. Given the near 100% accuracy around +1s after the sample-poke, I wonder if it would be worthwhile extending this analysis for several more seconds - i.e. across the delay - in order to answer the crucial question how long the to-be-memorized stimulus identity is encoded in each of these regions in their original representation (akin to delay activity). I acknowledge that this is very similar to the analysis presented in Figure 3G, but I think it might be worthwhile doing this analysis with an alignment to the sample-phase poke as well, to start with the original representation.

Also, for the decoding analysis in Fig. 2 and 3, it would be worthwhile stating explicitly if this has been done on the level of individual animals or at the level of the pseudopopulation (as if all neurons from one region would derive from one animal) only. In the latter case, it should be explained how this was done, given that for every actual choice made by an actual animal only a subset of neurons in the pseudopopulation (i.e. the neurons recorded in the brain of that animal) is actually "available" for the decoding analysis. The same question applies to the decoding analysis of figure 5C.

3. It is not clear to me from the task description, if trials could also include the center poke as sample-phase poke-hole. This design would render the task to be more like a delayed-choice (as opposed to delay-response) task, because in the design with either the left or the right sample-phase hole, mice "just" need to remember to go away from that hole towards the middle; albeit with the additional condition of the middle sample-phase hole such a simple encoding is not possible; mice might have to go either right or left in the choice phase. So if this latter option existed, it would be worthwhile investigating the choice accuracy on these trials (vs. the other ones) and possibly also the encoding.

4. Figure 3G: The authors conclude that it is largely the same dmPFC neural ensembles representing the to-be-encoded stimulus by more or less stable activity. This is a very important finding, and I guess not only the accuracy level along the diagonal line but also the spread of high accuracy levels across the two time axis (indicating that a classifier trained at any time point can be used to decode the maintained stimulus at any time point during the delay) could be mentioned as additional argument. I wonder though, what the

dispersion in the plot (decrease of accuracy, esp. along the diagonal line) reflects at the level of delay activity of individual neurons, since Fig. 5B-F suggest that even in dmPFC there are only few neurons with stable delay-activity. For example, one could remove such few neurons with prolonged increased (or decreased) activity across the whole delay and compare the decoder performance in order to see if decoding was based on just a few sparse "delay-cells" or if the stimulus can also be encoded by populations of cells that themselves do not show stable activity.

5. I am somewhat surprised about the very high CPD values presented in Figure 5A-B. In Akam et al (2021), which the authors themselves cite for this approach, CPDs of individual factors are no higher than 5% and summed up CPDs of all behavioural predictors typically do not exceed 20%. In that respect, I also do not understand Figure 5B. A CPD of 100% would imply that ALL variability found in the neural activity is explained by the - still relatively few - behavioural parameters that the authors chose a priori. This is biologically implausible, because there is always noise in neural activity and there are also other factors that likely influence neural activity but are not predictors in this analysis (e.g. motivational or attentional state, trial history, other motor and sensory variables, etc.). Likely the %values do not refer to CPD itself but to some other base. The authors should make explicit what this base is (i.e. % of what?) and what the actual predictor matrix behind the CPD analysis was. Likewise, the time axis in Figure 5 is confusing with respect to the other three factors given the difference between them; i.e. while for task-phase/context the time around the 3 pokes is used for differentiation (so "Poke" here refers to all 3 pokes at once), this is not the case for the other 3 factors, for which only the pie-chart is shown unrelated to time. The CPD-values for sample/choice poke location and outcome need to be shown over time in alignment to both the sample and the choice poke separately as is done in Figure 6; so the reference time point and general value of Figure 5B is not clear, in my view - Figure 5 A and Figure 6 give the accurate depiction of the encoding analysis. As stated above, it should also be made clear if these encoding and decoding analyses were done for each animal individually, and if not, how the combination of (neurons of all) animals was achieved in the analysis.

6. l. 32, it is arguable to what extent cognitive processing is hierarchical, rather than distributed (<https://www.nature.com/articles/nrn.2017.7>) and to what extent the PFC is really at the top of this hierarchy; considering different circuits being required for different tasks or types of WM and considering potential differences between species, this statement appears to be too general; for example, in humans, the PFC does not necessarily have a dominant role compared to other cortical areas

(<https://journals.plos.org/plosbiology/article?id=10.1371/journal.pbio.2004274>;
<https://www.biorxiv.org/content/10.1101/2021.04.20.440621v1>) and manipulations of the rodent PFC, incl. Cg1 (where the authors may be recording with their dmPFC electrodes) may have varying effects on behavioural performance, not more profound than modulation of MD thalamus or hippocampus (see e.g., <https://www.nature.com/articles/srep16778> ; <https://www.nature.com/articles/s41598-021-88200-z> ; <https://www.ncbi.nlm.nih.gov/pmc/articles/PMC5501395/> ; <https://www.ncbi.nlm.nih.gov/pmc/articles/PMC10939314/>).

7. Could the authors briefly comment on why striatum was used as reference electrode location? I guess for the analysis of action potentials it is o.k. but if one wanted to analyse LFP or connectivity information, the coupling between striatum and frontal areas would likely confound the analysis.

Reviewer #2 (Remarks to the Author):

The manuscript by Sonneborn et al ‘Divergent Subregional Information Processing in Mouse Prefrontal Cortex During Working Memory’ addresses an important and relevant topic – how are subregions of prefrontal cortex involved in information processing in a behavioral task. Mice were trained in a delayed non-match to place task, custom-made electrode bundles were used to record neural activity separately in MO, dmPFC and vmPFC. Modulation of neural activity by task phase and other variables was then analyzed and compared among subregions.

1. My main concern with the manuscript is a lack of behavioral analysis, without these it is difficult to draw conclusion about potential functional correlates of the observed activity. For instance, what was the performance of the animals during the task? The authors mention a threshold of 70% correct trials before the animals underwent surgery, but how about the performance during the recordings?

2. The recordings were not conducted simultaneously in different brain areas and comparisons are made across different sessions without information about behavioral parameters – how might running speed, differences in choice behavior, motivation, motor actions, .. affect the results? Is the observed modulation in neural activity consistent across session? Do all results hold up for different sessions? There is no information about the results from individual sessions.

3. My understanding of the task is as follows: during the sample phase left or right port is active, as indicated by a visual cue. During the test phase the previously lit port and either the port on the side or the center port is lit and the animal has to decide which one was not lit in the sample phase. However, the center port is never the sample, so in case it is lit (50 % of trials) during the test phase, it is always correct and the animal does not have to remember the sample location, which would make it a simple stimulus response task and not a working memory task for these trials. If animals followed that strategy and guessed the remaining 50 % of trials where the other possible sample port is lit, it would overall achieve a performance of over 70% without using its working memory. Please clarify. As the authors stress the 'working memory' aspect of the task and activity that is potentially related to it, this is central.

The authors should also split the trials according to type and show the success rate separately for the different types.

4. How many sessions with how many neurons in each brain area were recorded?

5. Page 5 line94: please justify your choice for this threshold

6. Fig2 c what is the shaded area? This should be mentioned in the legend

7. Fig 2d - is it possible that an overlapping population of neurons is first increasing and then decreasing, for example in vmPFC? Please also show how single neurons are modulated at least for a few time points, it's hard to see in the condensed form of 2B. It would be helpful to show Fig. 2B split by or organized according to recording session, as mentioned before and also for other figures

Is 2f simply the sum of the two lines of each color in 2d with different normalization? These slight variants with very similar message are a bit confusing, it could be combined or reduced to only one figure.

Fig2g is there an effect of session/speed of the animal/other behavioral factors?

8. Fig 3f should also show percentage of cells selective for the other location in the delay period (i.e not the sample) to get an idea on how specific this representation is.

9. Fig 3g - how about error trials? If the authors think dmPFC holds the information in the delay period specifically, they should show the same plot for the error trials. Again, please show information about how consistent this is across different sessions.

10. Some of the information from Fig 2-4 could be condensed, for example 3g and 4g
11. Fig. 5: isn't the delay in a different location and the 'context' is simply the location? For the left and right ports the comparison is more clear because here the same location can be visited as the sample or choice. Could any modulation also be an effect of the reward? Is the context the reward? The same question arises for outcome, which could also be a reflection of reward or no reward and not a cognitive variable like context – please clarify.
12. Fig 6B typo in legend 'detectible'
13. Page 8 line 172: 'we needed to remove the poke context task variable due to its dependence on a 172 prohibitively small window around pokes' – wouldn't the contribution of the other variables still be detectable?
14. Page 8 last paragraph, sentence appears not complete
15. Page 9, line 207 'Briefly, would found that the sample-selective subpopulations of MOs and dmPFC neurons have similar beta weight distributions when comparing sample and delay pokes, while the vmPFC subpopulation is negatively correlated.' Typo – another typo further down on the same page
16. Fig 7 add information on neuron numbers
17. Fig 8 critical information when the reward is delivered is missing
18. Discussion, p.12 : 'dynamic flow of information from MOs to vmPFC as the mice progress through the task' – without simultaneous recording and no information on the behavior in the task such a statement is not appropriate.
19. P.13 'Another challenge has been the difficulty in eliciting persistent WM activity in rodent PFC to be able to study the underlying mechanisms in mice. By identifying such activity, as well other types of WM activity, this study provides a foundation to perform such studies using tools that are uniquely available in mice.' – doesn't the fact that persistent activity can often not be observed demonstrate that in these tasks persistent activity cannot be the mechanism of working memory (or they are not working memory tasks)? Furthermore, these results demonstrate that any correlates have to be examined closely in the context of behavior and task specifics and be described in this context. Whatever the behavioral or cognitive correlate of the identified persistent activity in this manuscript is,

there is no evidence that it would be a general mechanism for working memory. This sentence ignores the lack of behavioral analysis (it is mentioned before though) and makes it into a selling point even though the nature of the correlate is unknown.

Reviewer #3 (Remarks to the Author):

In this interesting study the authors recorded units in different subregions of the prefrontal cortex (PFC) and directly compared their firing properties during the execution of working memory (WM) in a delayed non-match to sample paradigm. The main finding of this study is that, during WM delay, dorso-medial PFC (dmPFC) neurons maintain consistent firing patterns that contain information about which reward port served as the 'sample' in the current trial. This finding is complemented by observations on task-phase and outcome selectivity of neurons in the different regions. The findings are highly relevant to the field and are presented in a very clearly written and well-structured manuscript. Moreover, the data are clearly presented in the figures, which allows the reader to easily grasp the main findings.

While I fully support publication of this manuscript, I think that a number of minor (and mostly technical) points should be addressed to further strengthen the conclusions from this study:

1. SVM decoding (line 106, Fig. 2): This analysis could be further strengthened by comparing the decoder results to shuffled data (e.g., by using surrogate data for each pseudopopulation composed of spike trains for which the inter-spike intervals have been shuffled). This way, a proper statistical analysis could be performed. Moreover, it would be informative to see whether the decoding results hold if performed for each mouse separately. With that approach, fewer neurons would be available for the model, so we would expect overall lower decoding accuracy, but the authors could still evaluate those results against shuffled data. Finally, a larger number of MO neurons shows significant discrimination between the two sampling ports. Does that mean that significant decoding can be achieved with fewer neurons from that compared to the other regions? This could be tested by randomly subsampling from the pseudopopulation (or on a mouse-by-mouse basis) using increasing numbers of units.

2. Figs 3 and 4 show differential dmPFC activity during the delay and prior to the choice.

Since information about the sampling port visited earlier in the trial is the WM component of this task, this provides evidence for a persistent activity mechanism. However, the authors rightfully mention (in line 130) that a ‘temporal code’ has been proposed as an alternative mechanism to persistent activity. To strengthen the authors claim that persistent activity is the dominant mechanism in this task, it would be very interesting to thoroughly check the data for traces of temporal coding. For instance, would it be possible to decode trial outcome (or sampling port identity) during the delay based on temporal information only (e.g., from the time of peak activity of each neuron)? Or, when sorted by time of peak activity during the first delay period, do the neurons fire again in the same order during subsequent delays of the same trial type? The latter could be tested for significance based on correlations of the obtained temporal tuning functions compared to those of shuffled data.

3. On a more conceptual level: Fig. 3G shows that the decoding accuracy of dmPFC neurons drops substantially later in the delay, even when only considering the same training/testing time bins. Does this mean that information about which sample side has been visited earlier in the trial gradually disappears from the spiking of these neurons? If so, I’m puzzled how the information comes back, since Fig. 4G shows very high accuracy prior to the choice (when the animals already approach the choice port, I assume). It would be interesting to provide an interpretation of that observation in the discussion.

4. Fig. 5C (line 166). The multi-class SVM is very accurate with neurons of all three regions. Given that more of the MO cells’ variance in firing is explained by poke context (Fig. 5A), I wonder whether the relatively good performance of the other regions might be a saturation effect due to the number of neurons used for decoding. Again, analyzing accuracy as a function of randomly drawn neurons could help to identify more subtle difference between the regions.

5. Anatomy of recording sites: Would it be possible to state in which anatomical PFC subregions (e.g., piriform cortex, cingulate cortex) the individual recording sites are located? From the histology shown in Fig. 1 it is not clear whether all recording sites from each experimental group (e.g., dmPFC) fall within the same ‘classical’ anatomical region.

6. Pseudopopulation sizes: The authors state in line 80 that the pseudopopulations for the three regions differed in size. Did they control for this (by subsampling) in the SVM analyses? This should be clarified.

7. The task design used by the authors is quite elegant, and the fact the the rewarded port is

not known until after the delay is a clear strength of this task. However, I wonder whether trials with the two possible target sites differ from each other in terms of difficulty. Given the left port is the sample, as shown in the example in Fig. 1A, would a trial in which 'center' is the target be more difficult than a trial with 'right' as the target? After all, the non-match and sample sites are closer together in that case. Would it be possible to check the performance for both near and distant match sites separately to rule this out?

Dear Editor and Reviewers,

We appreciate the opportunity to revise our manuscript. We would like to thank the reviewers for their thoughtful comments on the manuscript and appreciate the excellent suggestions that were made. Below, we address each comment/suggestion/question one-by-one. We believe the resulting revised manuscript is substantially improved as a result.

Below, the reviewer comments are highlighted in green, and our responses are shown in blue text. In the responses, we describe the results of the new analyses and in some cases show them. The changes in the manuscript are highlighted in red within the manuscript, and there is a new Supplementary Figures file with eight new figures. Since there was such a large amount that was changed, we chose to not include most of the changes in the Response document so that the response would not be overcrowded.

Reviewers' comments:

Reviewer #1 (Remarks to the Author):

The authors present a highly interesting, excellently conducted and analysed study revealing key neural mechanisms of working memory (WM) in mice. They record single-cell ensembles in three frontal regions, the supplementary motor area (MO), the dorsomedial PFC (dmPFC; although according to coordinates that might rather be ACC/Cg1 or Cg1/PrL border) and the ventro-medial PFC (vmPFC, akin to classical IL). Importantly, they do so in an advanced, operant rodent WM-task, that (compared to commonly used, simple alternation-based maze-assays) is more similar to human WM tasks and specifically to delayed-choice (as opposed to delayed-response) tasks, which afford better control over task timing and other psychological variables (attention, motivation) and allow to better discriminate actual memory retention from preliminary motor-programming. In addition to introducing and using this very informative task, the authors also make important revelations regarding distinct roles of the investigated areas; most importantly they show that MO contains the strongest encoding of the stimulus-identity = spatial motor action (poke-hole) during and several seconds after poking, but only the dmPFC retains the to-be-memorized stimulus representation throughout all of the very long delay, functioning as the actual memory store - and seemingly largely based on persistent activity (see the

debate <https://www.jneurosci.org/content/38/32/7020> vs. <https://www.jneurosci.org/content/38/32/7013> for reference). During choice, vmPFC (in addition to the other areas) also becomes highly active and encodes the to-be chosen poke-hole, which is also a discrimination between rewarded and non-rewarded choice. Therefore, one could imagine a functional distinction between such areas as MO - as motor area - encoding the spatial determinants of the motor action before, during and after poking-action, dmPFC (or Cg1) encoding the actual memory in absence of motor/sensory information, and vmPFC possibly encoding reward-related aspects.

I have no major reservations against publication of this excellent manuscript in Communications Biology, but have a few minor suggestions:

1. Figure 2: It could be worthwhile having some statistical measure for the the share of neurons

modulated by an event in each of the three brain regions, to know if some neurons can be identified as activated or inhibited during the sample-phase poke with some statistical certainty. It is unclear to me, to what extent the analysis in 2D contains such statistical certainty and what the actual comparison interval for the 0.5 Z-unit increase is and what the p-value stated in I. 98 and indicated in 2D actually refers to. E.g. one could use a within-subject Wilcoxon-test across trials comparing amplitudes in a baseline interval to a certain time window around the poke (<http://www.sciencedirect.com/science/article/pii/S0092867417312047>). Using a Chi-square test comparing the share of activated/inhibited/unchanged neurons one could get a second indication for the statement that MO differs from the two PFC regions, but not the latter from each other.

Reply: This is a good suggestion, and another reviewer also had a similar concern. Our initial decision to use 0.5 Z-units as a cutoff was arbitrary. We have re-analyzed and replaced all subfigures in question (2d, 3d, and 4d) using a Wilcoxon signed-rank test comparing all important time points to the mean, on each trial, of 1 second of activity near the end of a baseline period in the intertrial interval (ITI). We then ran our same Chi-square analysis to determine differences in proportions. We ended up finding many more significantly modulated neurons this way, and we think it is a better representation of the population Z-scored firing rates in Figs. 2b, 3b, and 4b.

As per the other reviewer, we have also added Supplementary Fig. 2, depicting Venn diagrams containing the percent of these neurons in each region that significantly increase, decrease, or **both increase and decrease** their firing rates in response to important task variables or around important task events. A majority of neurons seemed to display a significant response (increase or decrease or both) in at least two 100 ms bins around all important events. Supplementary Fig. 2b also has some example Z-scored traces from individual neurons that exhibited **both increased and decreased** firing from baseline.

2. Figure 2G. The cross-temporal SVM is a very interesting and revealing approach. Given the near 100% accuracy around +1s after the sample-poke, I wonder if it would be worthwhile extending this analysis for several more seconds - i.e. across the delay - in order to answer the crucial question how long the to-be-memorized stimulus identity is encoded in each of these regions in their original representation (akin to delay activity). I acknowledge that this is very similar to the analysis presented in Figure 3G, but I think it might be worthwhile doing this analysis with an alignment to the sample-phase poke as well, to start with the original representation.

Reply: As per another reviewer, we have redone this analysis with shuffled controls (shuffling right and left sample port identity) in order to determine statistical decoding significance in each train/test time bin. All reviewers asked for some form of shuffled SVM control to determine this significance. Since the shuffling needs to be done many times (we did 1000) on each train/test decoding bin, we increased the time bin length to 200 ms to help with the large increases in computation time that come with shuffling. To our surprise, this widening of the time bins substantially increased decoding accuracy of many cross-temporal time points.

As suggested, we have also extended the time of decoding from 1 second to 5 seconds (Figure 2g) after the sample poke, which shows that the dmPFC has the most stable selective activity from the sample poke extending into the delay period.

Also, for the decoding analysis in Fig. 2 and 3, it would be worthwhile stating explicitly if this has been done on the level of individual animals or at the level of the pseudopopulation (as if all neurons from one region would derive from one animal) only. In the latter case, it should be explained how this was done, given that for every actual choice made by an actual animal only a subset of neurons in the pseudopopulation (i.e. the neurons recorded in the brain of that animal) is actually "available" for the decoding analysis. The same question applies to the decoding analysis of figure 5C.

Reply: The analyses for the main SVM figures were done on the level of the pseudopopulation, and we have added additional information about this into the Methods section. The rationale for using pseudopopulations is based on a large number of recently published approaches:

1. <https://www.pnas.org/doi/full/10.1073/pnas.1619449114>
2. <https://www.jneurosci.org/content/44/6/e0703232023.full>
3. [https://www.cell.com/neuron/fulltext/S0896-6273\(24\)00047-3](https://www.cell.com/neuron/fulltext/S0896-6273(24)00047-3)
4. <https://www.jneurosci.org/content/43/25/4650.full>
5. <https://nature.com/articles/s41593-023-01472-8>
6. <https://www.nature.com/articles/s41593-023-01461-x>

Moreover, the pseudopopulation approach is consistent with our own observations that different animals in the same implant location groups tended to display comparable neural activity around all the main task points (pokes, delay hold, and outcome). This likely has to do with the phenomenon that salient events, such as our poke events, can trigger robust, reproducible, low-dimensional representations across the brain. In this study, some of these representations seemed to be shared across subregions, while others were unique.

In relation to this question, another reviewer also asked about session-based electrophysiological analysis. We have included 3 new Supplementary Figures (Supp Figs. 5, 6, and 7) showing the sample-identity SVM analysis during delay broken down by individual sessions.

3. It is not clear to me from the task description, if trials could also include the center poke as sample-phase poke-hole. This design would render the task to be more like a delayed-choice (as opposed to delay-response) task, because in the design with either the left or the right sample-phase hole, mice "just" need to remember to go away from that hole towards the middle; albeit with the additional condition of the middle sample-phase hole such a simple encoding is not possible; mice might have to go either right or left in the choice phase. So if this latter option existed, it would be worthwhile investigating the choice accuracy on these trials (vs. the other ones) and possibly also the encoding.

Reply: We apologize for the confusion here; the center port was not used as a sample location for these experiments. We have updated the Methods section and legend in Figure 1 to more clearly state this. We agree that the task design could potentially turn center choice trials into more of a delayed stimulus-response task. But as you point out, we believe that the mice still need some form of working memory during the delay since they cannot know which choice port (center or outer) will light up until after the delay ends. This may be more of a motor plan working memory ("I am planning to turn away from the sample port after the delay ends") than

the canonical holding of the old location in mind and saving the decision about where to go until after the delay (canonical sensory working memory). We unfortunately do not have a good way to tease these possibilities apart in this current dataset.

With respect to your final point, we have added some more detailed behavioral analyses in Supplementary Figure 1 showing that mice perform at 87.6% on trials where the center port is the choice, which is lower than the ~100% performance expected if the mouse was using a pure delayed stimulus-response strategy. Furthermore, they perform at 75% on average on outer trials, much greater than the 50% expected if they were using the delayed stimulus-response center port strategy on center trials and just guessing on the outer trials. The rest of this figure also contains evidence that mice generally do not seem to be treating the center port differently than the outer ones.

4. Figure 3G: The authors conclude that it is largely the same dmPFC neural ensembles representing the to-be-encoded stimulus by more or less stable activity. This is a very important finding, and I guess not only the accuracy level along the diagonal line but also the spread of high accuracy levels across the two time axis (indicating that a classifier trained at any time point can be used to decode the maintained stimulus at any time point during the delay) could be mentioned as additional argument. I wonder though, what the dispersion in the plot (decrease of accuracy, esp. along the diagonal line) reflects at the level of delay activity of individual neurons, since Fig. 5B-F suggest that even in dmPFC there are only few neurons with stable delay-activity. For example, one could remove such few neurons with prolonged increased (or decreased) activity across the whole delay and compare the decoder performance in order to see if decoding was based on just a few sparse "delay-cells" or if the stimulus can also be encoded by populations of cells that themselves do not show stable activity.

This is an interesting idea. We have added Supplementary Figure 4, which shows what the same dmPFC SVM decoding from Fig. 3g looks like after removing 29 out of the 354 neurons from the dmPFC which displayed at least 20 bins (2 total seconds) of significant coding for sample location during the delay. We have added a description of this to the main text of the manuscript. We have also updated our language to include more words like "on- and off-diagonal" as suggested. Here are the two analyses next to each other for comparison.

5. I am somewhat surprised about the very high CPD values presented in Figure 5A-B. In Akam et al (2021), which the authors themselves cite for this approach, CPDs of individual factors are

no higher than 5% and summed up CPDs of all behavioural predictors typically do not exceed 20%. In that respect, I also do not understand Figure 5B. A CPD of 100% would imply that ALL variability found in the neural activity is explained by the - still relatively few - behavioural parameters that the authors chose a priori. This is biologically implausible, because there is always noise in neural activity and there are also other factors that likely influence neural activity but are not predictors in this analysis (e.g. motivational or attentional state, trial history, other motor and sensory variables, etc.). Likely the %values do not refer to CPD itself but to some other base. The authors should make explicit what this base is (i.e. % of what?) and what the actual predictor matrix behind the CPD analysis was.

Reply: We were also initially surprised by our high CPD values, but there may be several plausible explanations for why they are higher:

- 1) The Akam paper uses calcium imaging as opposed to electrophysiology, which may reduce the ability to temporally resolve smaller differences in firing rates between regressor levels.
- 2) Relatedly, a direct quote from that paper states that “Activity was sparse, with an average event rate of 0.12 Hz across the recorded population (Figure 3C).” Basically, they recorded a **very** low calcium event rate on average during behavior, and they did not have **any** cells above 0.4 Hz according to their histogram. This average seems extremely low and was not the case for our single units. We actually did not include any neuron in our analyses with an average firing rate across an entire session below 0.5 Hz (also added to the main text). Thus, they may not have been capturing the same temporal aspects of the data as we did.
- 3) The Akam study also included interaction terms in their predictor matrix, while we did not, as we wanted to look at each variable individually. In a GLM, the more variables you use in an attempt to explain firing rate variability, the weaker any given variable's contribution to the overall firing rate variability will be. This is especially true if the extra variables are interaction terms containing the original variable.

Furthermore, other studies calculating coefficients of partial determination (CPDs) for variables analogous to ours using GLMs have also found CPD values much higher than the Akam paper. See Hocker et al. (<https://elifesciences.org/articles/70129>) for CPD values up to 15% in Figure 4A.

Overall, the main reason we used the Akam study as a reference was because we thought their time-dependent GLM method was the best suited method for our dataset and the questions we wanted to answer, and we did not necessarily expect to get similar results. Either way, regardless of the CPD magnitude for any individual variables, there was still a large amount of unexplained variance with our model in all subregions (Fig. 5a).

For the last point about the predictor matrix, we have added the following more detailed description about dummy variable creation and using categorical predictors to the methods.

“Construction of the GLM predictor matrix was done by converting the levels of these regressors (for example, left or right for sample location) into dummy variables for each neuron (MATLAB function *dummyvar*, in the above example, left sample trials become 1 and right

sample trials become 0 for an individual session). When training a GLM model to predict neural firing rate, each of these one-hot encoders was treated as a categorical predictor variable. In the first GLM analysis examining encoding of task phase context, the three predictor matrices for task phase were stacked on top of each other such that poke time was collapsed. This contrasts with the second GLM analysis examining encoding of the other regressors which could be done across the entire trial (as opposed to just around the poke). Since we were doing this for individual neurons, the length of each stacked predictor matrix varied, and was equal to the number of trials in a given neuron's session multiplied by 3 (for the 3 stacked task phase matrices)."

Likewise, the time axis in Figure 5 is confusing with respect to the other three factors given the difference between them; i.e. while for task-phase/context the time around the 3 pokes is used for differentiation (so "Poke" here refers to all 3 pokes at once), this is not the case for the other 3 factors, for which only the pie-chart is shown unrelated to time. The CPD-values for sample/choice poke location and outcome need to be shown over time in alignment to both the sample and the choice poke separately as is done in Figure 6; so the reference time point and general value of Figure 5B is not clear, in my view - Figure 5A and Figure 6 give the accurate depiction of the encoding analysis. As stated above, it should also be made clear if these encoding and decoding analyses were done for each animal individually, and if not, how the combination of (neurons of all) animals was achieved in the analysis.

Reply: Sorry for the confusion, we have carefully re-worked this figure and its legend, along with its description in the main text to hopefully make this analysis much clearer. To be sure we are on the same page, this GLM analysis was always run on each *neuron* in each pseudopopulation individually, and then averaged across the whole pseudopopulation to get the mean contribution of our task variables to pseudopopulation firing rate variability. Two GLM analyses were performed (see description in previous response), one aimed at differentiating task phase across pokes and the other aimed at examining encoding of the other regressors across entire trials. Changes to text below (and in the revised manuscript) are in red.

Fig. 5: DNMT task variables account for the most poke-related firing rate variability in MOs. **a** The coefficient of partial determination (CPD, calculated for each neuron) from a general linear model was used to estimate the proportion of total pseudopopulation firing rate variance around pokes that was either unexplained (white bars) or that could be explained by our four regressors (Poke Context, Sample Port, Choice Port, and Outcome). Numbers under each regressor represent the mean CPD (% variance explained) across five 200 ms time bins around the poke for that regressor (this time frame can be visualized in the following panel, 5b). A one-way ANOVA revealed a significant main effect of subregion on explainable variance, $F(2, 985) = 44.54, p = 3.02e-19$. Bonferroni post-hoc comparisons found that the MOs had the most explainable variance among subregions, while vmPFC contained the least (black asterisks, MOs vs. dmPFC $p = 3.02e-7$, MOs vs. vmPFC $p = 8.78e-20$, dmPFC vs. vmPFC, $p = 5.35e-5$). Furthermore, in each subregion, one-way ANOVAs with Bonferroni post-hoc tests found that of the four regressors, Poke Context (PC) was the largest contributor to explained variability. Colored asterisks represent significant differences within subregions between the CPD values for Poke Context and all three other regressors. **b** Looking at the CPD of Poke Context in isolation at all time points around the poke, we confirm that this regressor accounts for significantly more firing rate variability in the MOs at all time points around pokes, followed by dmPFC and then vmPFC. Double-colored straight lines represent statistically significant differences (p -value < .05) between the two respective subregions in that time bin, after correcting for both false discovery rate and family-wise error rate (see **Methods**). Lighter shaded areas above and below the solid lines represent the standard error of the mean CPD from the pseudopopulation at each time point. **c** Despite MOs containing the highest CPD for Poke Context, Poke Context is decodable with nearly 100% accuracy in all regions around pokes using a linear support vector machine (SVM). Chance level decoding in the shuffled control (dashed lines) is 1/3. **d** Subsampled SVMs revealed that the MOs can decode Poke Context with fewer cells than the other subregions.

6. I. 32, it is arguable to what extent cognitive processing is hierarchical, rather than distributed (<https://www.nature.com/articles/nrn2017.7>) and to what extent the PFC is really at the top of this hierarchy; considering different circuits being required for different tasks or types of WM and considering potential differences between species, this statement appears to be too general; for example, in humans, the PFC does not necessarily have a dominant role compared to other cortical areas

(<https://journals.plos.org/plosbiology/article?id=10.1371/journal.pbio.2004274>; <https://www.biorxiv.org/content/10.1101/2021.04.20.440621v1>) and manipulations of the rodent PFC, incl. Cg1 (where the authors may be recording with their dmPFC electrodes) may have varying effects on behavioural performance, not more profound than modulation of MD thalamus or hippocampus (see e.g., <https://www.nature.com/articles/srep16778>; <https://www.nature.com/articles/s41598->

021-88200

z ; <https://www.ncbi.nlm.nih.gov/pmc/articles/PMC5501395/> ; <https://www.ncbi.nlm.nih.gov/pmc/articles/PMC10939314/>).

Reply: We agree that our statement about this phenomenon was too strong and have amended it to better acknowledge the distributed nature of WM in the first paragraph of the Introduction.

7. Could the authors briefly comment on why striatum was used as reference electrode location? I guess for the analysis of action potentials it is o.k. but if one wanted to analyse LFP or connectivity information, the coupling between striatum and frontal areas would likely confound the analysis.

Reply: We have empirically found that we see less noise using “in-brain” referencing and have selected striatal electrode instead of the more commonly used reference screw over the olfactory bulb for better impedance matching (and therefore better referencing), thereby enhancing our neuronal yield, and because the striatum receives prominent volume conducted signals, including at least some of those that reach cortical areas (see <https://www.ncbi.nlm.nih.gov/pmc/articles/PMC5616191/>). Striatum has the added advantage of being a large target that can easily be hit consistently. We agree that we will need to carefully reconsider the referencing scheme in any project that is more LFP focused.

Reviewer #2 (Remarks to the Author):

The manuscript by Sonneborn et al 'Divergent Subregional Information Processing in Mouse Prefrontal Cortex During Working Memory' addresses an important and relevant topic – how are subregions of prefrontal cortex involved in information processing in a behavioral task. Mice were trained in a delayed non-match to place task, custom-made electrode bundles were used to record neural activity separately in MO, dmPFC and vmPFC. Modulation of neural activity by task phase and other variables was then analyzed and compared among subregions.

1. My main concern with the manuscript is a lack of behavioral analysis, without these it is difficult to draw conclusion about potential functional correlates of the observed activity. For instance, what was the performance of the animals during the task? The authors mention a threshold of 70% correct trials before the animals underwent surgery, but how about the performance during the recordings?

Reply: We apologize that this was not clear in the original manuscript, but the analysis in Figure 1c only contains behavioral sessions *after* electrode implantation and with simultaneous single-unit recordings. We have updated the Results section (paragraph 1) and Figure 1 legend to make this very clear.

2. The recordings were not conducted simultaneously in different brain areas and comparisons are made across different sessions without information about behavioral parameters – how might running speed, differences in choice behavior, motivation, motor actions, affect the results? Is the observed modulation in neural activity consistent across session? Do all results hold up for different sessions? There is no information about the results from individual sessions.

Reply: Unfortunately, we did not collect data on running speed, motivation, or motor actions in this study (only indirectly via the time it takes mice to travel from poke to poke), so some analyses are inaccessible. However, we broke down the behavior into some more in-depth analyses containing data from individual sessions (new Supplementary Figure 1). See the response to the comment directly below this one (“3.”) for a more pertinent and detailed explanation of those analyses.

Due to both computational limitations and spatial limitations for figures, we could not break down every single analysis into individual sessions. However, we suspected that the delay SVM analyses (Fig. 3g) would be the most interesting to look at when looked at on a session-by-session basis. Thus, we performed session-based cross-temporal SVMs that were similar to our delay SVMs for the MOs, dmPFC, and vmPFC pseudopopulations in Fig. 3g.

We report several interesting results from this analysis in three new Supplementary Figures (Supp. Figs. 5, 6, and 7). The most relevant finding to the current and the following reviewer comments was the fact that in the dmPFC, 5 out of the 6 mice had at least one session with substantial cross-temporal decoding of sample identity throughout the delay. Three of the mice also had more than one session with strong and stable decoding. Moreover, the one mouse (dmPFC 4) which did not have any sessions with cross-temporal decoding was the same mouse we identified in the supplemental behavioral analysis (Supp Fig. 1) to possibly not be doing working memory. This finding, along with the behavioral analyses described below, provides

strong evidence that most of the mice are doing working memory, and that the neural activity we see during the delay is related to working memory processes in the medial prefrontal cortex. The other two subregions did not seem contain any sessions with strong and stable decoding capability.

Overall, we believe that using pseudopopulations should not affect the main interpretation of our findings. The rationale for using pseudopopulations and comparing across brain regions without recording simultaneously is based on a large number of studies. Links to a number of recently published articles using this approach are displayed below. We do agree that a fruitful future direction would be to recording simultaneously across these brain regions.

1. <https://www.pnas.org/doi/full/10.1073/pnas.1619449114>
2. <https://www.jneurosci.org/content/44/6/e0703232023.full>
3. [https://www.cell.com/neuron/fulltext/S0896-6273\(24\)00047-3](https://www.cell.com/neuron/fulltext/S0896-6273(24)00047-3)
4. <https://www.jneurosci.org/content/43/25/4650.full>
5. <https://nature.com/articles/s41593-023-01472-8>
6. <https://www.nature.com/articles/s41593-023-01461-x>

3. My understanding of the task is as follows: during the sample phase left or right port is active, as indicated by a visual cue. During the test phase the previously lit port and either the port on the side or the center port is lit and the animal has to decide which one was not lit in the sample phase. However, the center port is never the sample, so in case it is lit (50 % of trials) during the test phase, it is always correct and the animal does not have to remember the sample location, which would make it a simple stimulus response task and not a working memory task for these trials. If animals followed that strategy and guessed the remaining 50 % of trials where the other possible sample port is lit, it would overall achieve a performance of over 70% without using its working memory. Please clarify. As the authors stress the 'working memory' aspect of the task and activity that is potentially related to it, this is central. The authors should also split the trials according to type and show the success rate separately for the different types.

Reply: This is an important point; your understanding is correct that the center port was not used as a sample location for these experiments. We have updated the Methods section, main text, and legend in Figure 1 to more clearly state this.

The bulk of this comment is concerned with the fact that the mice may not actually be doing working memory. We agree that the task design could potentially turn center choice trials into more of a delayed stimulus-response task. However, we believe that our additional behavioral analysis provides evidence that only one mouse is *potentially* using a simple delayed stimulus-response strategy, and the rest are likely doing the task how we would expect (Supplementary Figure 1). In Supp Fig. 1a We split the trials into choice-location-specific groups and show that mice perform at 87.6% when the choice is in the center, which is lower than the ~100% performance as you would expect from a pure delayed stimulus-response strategy. They also perform at 75% on average on outer trials, much greater than the 50% expected if they were just using the delayed stimulus-response strategy on center trials and guessing on outer trials.

Interestingly, the performance on center trials is significantly better than the outer ones, suggesting that some mice *may* be aware of this “center is always correct” rule. Ultimately, we don’t have a good way to definitively determine the source of this increased center performance. However, in Supp Fig. 1b, we can see that only in a small fraction of the sessions (within the black circle) did the mice perform very well in the center trials and ~50% on combined outer trials. Importantly, all of the dmPFC sessions (green dots) with ~50% outer trial accuracy were from the same mouse, with one outlier from a vmPFC session (pink dot). The rest of the sessions displayed a positive correlation between center and outer trial performance, indicating a spectrum of general engagement in the task. If all the animals were using a stimulus-response strategy, then you would see a much larger cluster at the top left of the Supp Fig. 1b graph.

As an alternative explanation for why mice perform better on center trials, it is possible that the mice simply had double the “practice” on center choice trials than they did on left or right ones. Intriguingly, the standard deviation of left (.161) and right (.189) trials is about double that of center trial performance (.083), indicating they may have honed their performance more on center trials.

We also reasoned that if mice were primarily using a stimulus-response strategy at the center port, then they would have a faster reaction time to the center choice port. This was not the case (Supp Fig 1c). The time to the choice poke on center vs outer trials has a significant positive correlation when plotted for individual sessions (Supp Fig. 1d), which suggests a spectrum of task engagement effect rather than a passive stimulus-response with guessing. This also argues against even the outlier group from Supp Fig. 1b using a stimulus-response strategy, and hints at them using another possible strategy, possibly with some sort of left or right side bias. Exploring the neural basis of different strategies is an interesting future direction.

Thus, we argue that the mice are likely using some form of working memory in this task since they cannot know which choice port (center or outer) will light up until after the delay ends. This may not be the typical retrospective sensory working memory often talked about in primates. In fact, it might actually be easier for mice to use more of a prospective motor-plan strategy to complete this task. For example, throughout the delay they wouldn’t need to remember the potentially more difficult retrospective sensory trace of the location they just came from, but instead they could prospectively hold the future correct turn direction or motor plan in mind while their nose is in the port.

4. How many sessions with how many neurons in each brain area were recorded?

Reply: We again apologize for the confusion on this point and have clarified this in the first paragraph of the Results section, as well as in the Figure 1 legend.

5. Page 5 line94: please justify your choice for this threshold

Reply: A similar concern was also brought up by another reviewer. Our initial decision to use 0.5 Z-units as a cutoff was arbitrary. We have re-analyzed and replaced all subfigures in question (2d, 3d, and 4d) using a Wilcoxon signed-rank test comparing all important time points to the mean, on each trial, of 1 second of baseline activity near the end of the intertrial interval period. We then ran our same Chi-square analysis to determine differences in proportions. We ended up identifying many more significantly modulated neurons this way, and we think it is a better representation of the population Z-scored firing rates in Figs. 2b, 3b, and 4b.

6. Fig2 c what is the shaded area? This should be mentioned in the legend

Reply: The shaded area for the line graph is the standard error of the mean at each time point. This is now in all applicable legends.

7. Fig 2d - is it possible that an overlapping population of neurons is first increasing and then decreasing, for example in vmPFC? Please also show how single neurons are modulated at least for a few time points, it's hard to see in the condensed form of 2B.

Reply: This is an interesting idea. We have added Supplementary Figure 2, depicting Venn diagrams containing the percent of these neurons in each region that significantly increase, decrease, or **both increase and decrease** their firing rates in response to important task variables. A majority of neurons seemed to display a significant response (increase or decrease or both) in at least two 100 ms time bins around all important events. The figure also has some example Z-scored traces from individual neurons which exhibited **both increased and decreased** firing from baseline.

It would be helpful to show Fig. 2B split by or organized according to recording session, as mentioned before and also for other figures

Reply: Splitting the heat maps by recording session is prohibitive due to the number of sessions and regions, which means many dozens of plots would be required. However, we have performed a large number of session-based analyses which we believe address the core idea here, which is to better understand how using session-level data affects the results. Please see Supplementary Figures 1, 5, 6, and 7. We also report interesting subsampling analyses in Supplementary Figures 3 and 4.

Is 2f simply the sum of the two lines of each color in 2d with different normalization? These slight variants with very similar message are a bit confusing, it could be combined or reduced to only one figure.

Reply: No, it is not. Figure 2f (and 3f and 4f) represents the percentage of sample-location-selective neurons over time while Figure 2d (and 3d and 4d) is the general increase or decrease in firing rate around task events, across all trial types regardless of where they went during the sample phase. There are many neurons that increase or decrease their firing rate without being selective, and also neurons that are selective which do not change their average firing rate (due to them increasing in half the trials (left) and decreasing in the other half (right), which can cancel out the firing rate when you average across all trials).

Fig2g is there an effect of session/speed of the animal/other behavioral factors?

Reply: Since we do not have a way to measure speed in this data set (we began recording video after this data set), we used the time from leaving the sample port to entering the delay port to estimate speed indirectly. In the figure below we plotted the mean decoding accuracy against the time it took mice to get from the sample port to the delay poke for each session. Only the MOs exhibited anything approaching a correlation ($r = -.41$), although it was not significant ($p = .07$). For dmPFC $r = -0.21$, $p = 0.33$; and for vmPFC, $r = 0.29$, $p = 0.14$.

8. Fig 3f should also show percentage of cells selective for the other location in the delay period (i.e not the sample) to get an idea on how specific this representation is.

Reply: Sorry about the confusion here, this figure is showing selectivity for *either* sample location. In other words, we plotted the combined percentage of neurons that had selectivity for either the left or right sample port location. Both sides were well represented in the population.

9. Fig 3g - how about error trials? If the authors think dmPFC holds the information in the delay period specifically, they should show the same plot for the error trials. Again, please show information about how consistent this is across different sessions.

Reply: This is an important addition to the paper. We have added Supplementary Figure 8 (shown below), and the following sentences in the main text:

“Finally, to confirm that the mice were holding sample identity information in mind during the delay, we performed a similar SVM analysis to compare correct versus incorrect trials. Using subsampling to account for the small number of incorrect trials per session (see Supplementary Fig 8 for details), we were unable to decode the sample port from neural activity in the periods leading up to or during the delay on incorrect trials (Supplementary Fig. 8). This suggests that performance on incorrect trials was not just a result of a lapse during the delay holding period, but was potentially due to lapsed encoding of the sample port before the delay even started.”

Supplementary Fig. 8: Cross-temporal support vector machine decoding of dmPFC sample identity during the delay on correct versus incorrect trials.

Similar to the delay decoding from Fig. 3g, cross-temporal SVMs were used to compare the ability to decode sample identity on correct versus incorrect trials. Importantly, since there were sessions without many incorrect trials and some mice had mild side biases, we only used sessions with at least 5 incorrect trials when the sample was on the left and 5 incorrect trials when the sample was on the right. This reduced the number of neurons used for this analysis to 180 (from 13 sessions), about half of the initial dmPFC pseudopopulation. **a** We then subsampled 5 left and right correct and incorrect trials 20 times, and took the mean decoding accuracy of these 20 subsampled SVMs at each cross-temporal bin to produce the heat maps shown. Note that since we have half the neurons and many fewer trials than for the dmPFC decoding in Fig. 3g, the overall decoding accuracy on Correct Trials is expected to be lower than that of the full pseudopopulation **b** Overall, we found significantly stronger sample identity decoding on Correct Trials, both before the delay poke and during the delay (paired t-test). Each black circle on these graphs represents the mean decoding in the time window 2 s before (**left**), or during the 5 second delay (**right**) for each of the 20 subsampled SVMs.

10. Some of the information from Fig 2-4 could be condensed, for example 3g and 4g

Reply: We have condensed panels in Figures 2-4 where possible.

11. Fig. 5: isn't the delay in a different location and the 'context' is simply the location? For the left and right ports the comparison is more clear because here the same location can be visited as the sample or choice.

Reply: You bring up a good point that a significant proportion of any spatial 'context' should be related to location. The reason we chose to call it 'context', and not something more specific like

'location', is that in our particular case 'context' is likely the combination of many factors that we did not directly collect data for in this paper. These could include location in the box, the direction the mouse turned before poking, understanding how much reward a port may give, motivation, nuances in a mouse's strategy, and many others. We therefore lumped all these together into the 'context' regressor since we could not separate them in any meaningful way.

Regardless of which variables are captured within the 'context' regressor, we can use SVM decoding to test if the pseudopopulations represent Left Sample pokes the same as Left Choice pokes, which would be expected if location was the only variable contributing to the larger 'context' designation. In contrast, we found we could easily decode Sample vs Choice Left pokes with nearly 100% accuracy all along the diagonal (trained and tested on the same time points) in all regions. Off-diagonal decoding seems to develop a wider window on poke approach as we move from MOs to vmPFC. These results provide evidence that the mice represent the Left port differently depending on the phase of the task. Likewise, in Figure 3g, the fact that we can decode the Sample location before, during, and after the delay poke means that the mice treat the back delay port differently when approached from the left versus the right, indicating they might be representing the back delay port in a more nuanced context than location alone.

Could any modulation also be an effect of the reward? Is the context the reward? The same question arises for outcome, which could also be a reflection of reward or no reward and not a cognitive variable like context – please clarify.

Reply: Related to the previous part of this question, it is highly plausible that reward contributes to context, but there is no way to definitively determine the extent to which it does. Indeed, the mice are likely aware that the sample poke gives no reward, the delay poke only gives a small reward after holding, and the last poke gives a large reward. We further argue that this is a function of their contextual understanding of each poke and falls under the 'context' regressor.

The Outcome variable is by definition accounting for whether or not the mouse got a trial correct or incorrect (i.e. received reward or not). Thus, we believe it is likely a combination of first realizing the choice was correct or incorrect, and then probably a difference in motor output due to the mouse licking or not. We included it in our model because it is one of the variables we definitively know the levels of the predictor variables for. It is also useful to include because in Figure 6c, we could try to predict whether the mouse would get the trial correct or not before the choice, which would provide insight into whether they might be "cheating" in some way. This did not seem to be the case. Interestingly, the MOs is the most responsive to the Outcome variable

after the choice, probably due to the mouse licking or not when water is dispensed or withheld. However, the ability to predict this variable does not peak until about 700 ms after the choice poke, which was outside the window for our GLM in Figure 5. Moreover, in Figure 8, the vmPFC seems to register the Outcome within 300-400 ms, which may be the mice actually realizing the absence or presence of the reward versus the action of licking/not licking and is within the time range of our GLM for Figure 5.

12. Fig 6B typo in legend 'detectible'

Reply: Thank you for catching this. This has been corrected.

13. Page 8 line 172: 'we needed to remove the poke context task variable due to its dependence on a prohibitively small window around pokes' – wouldn't the contribution of the other variables still be detectable?

Reply: The other variables would still be detectable. However, since we had to collapse the three poke contexts so that we could compare them in a single GLM (Fig. 5), we could not keep them collapsed in time if we wanted to run GLMs over time across the 3 separate phases (Fig. 6). Thus, we had to separate them as it is only possible to get 1 poke time course with them stacked, versus 3 separate phase time courses with them unstacked. We have added quite a bit of clarifying language to the methods, main text, and figure legends in the hopes of better explaining this rationale.

14. Page 8 last paragraph, sentence appears not complete

Reply: This has also been fixed – thank you for catching.

15. Page 9, line 207 'Briefly, would found that the sample-selective subpopulations of MOs and dmPFC neurons have similar beta weight distributions when comparing sample and delay pokes, while the vmPFC subpopulation is negatively correlated.' Typo – another typo further down on the same page

Reply: Thank you. Changed 'would' to 'we'. Also, removed 'is' further down.

16. Fig 7 add information on neuron numbers

Reply: We have added neuron number information into all subfigures and made this clearer in the main text.

17. Fig 8 critical information when the reward is delivered is missing

Reply: We have added a sentence to the legend and also color-coded the Choice/Reward time point to illustrate that the reward should occur (if the choice is correct) as they poke in the Choice port.

18. Discussion, p.12 : 'dynamic flow of information from MOs to vmPFC as the mice progress through the task' – without simultaneous recording and no information on the behavior in the task such a statement is not appropriate.

Reply: We have toned down our language here and worded it to be more of a future direction.

19. P.13 'Another challenge has been the difficulty in eliciting persistent WM activity in rodent

PFC to be able to study the underlying mechanisms in mice. By identifying such activity, as well other types of WM activity, this study provides a foundation to perform such studies using tools that are uniquely available in mice.' – doesn't the fact that persistent activity can often not be observed demonstrate that in these tasks persistent activity cannot be the mechanism of working memory (or they are not working memory tasks)?

Reply: It is a fair point that the question of the extent to which persistent activity is a WM mechanism is an unsettled one, although widely held views include that it is the primary or only mechanism, or that it is a special case with WM potentially stored in multiple formats that include persistent activity as one possible format (see dueling perspectives in <https://www.jneurosci.org/content/38/32> and work by the late Mark Stokes). We have revised the language to emphasize that the persistent neural activity we identified occurs during a WM task, and mouse tools would be useful to study its mechanisms. The wording is thus more agnostic as to the significance of the neural activity (WM or not), but still emphasizes the importance of studying the underlying mechanisms of a pattern of activity that occurs during at least some WM tasks and has been of interest to WM researchers for decades. The hope would that continuing more sophisticated studies across different types of WM tasks would help weigh on whether persistent activity is in fact a WM mechanism, whether it is general or a special case for some but not all WM, and whether WM is stored in multiple formats (less of a focus in this study but of interest to our group moving forward). As far as the question of whether rodent WM tasks in which persistent activity has not been found are in fact WM tasks, we agree that the idea that they are not (or perhaps are a special case that does not require persistent activity for reasons that are unclear) is one possible interpretation. Another point to emphasize is that few studies record more dorsally, especially in freely-moving rodents. Many are in fact in the vmPFC, where we see no such activity, so the fact that many prior studies have not found persistent activity may be related to recording location.

Furthermore, these results demonstrate that any correlates have to be examined closely in the context of behavior and task specifics and be described in this context. Whatever the behavioral or cognitive correlate of the identified persistent activity in this manuscript is, there is no evidence that it would be a general mechanism for working memory. This sentence ignores the lack of behavioral analysis (it is mentioned before though) and makes it into a selling point even though the nature of the correlate is unknown.

Reply: With the large amount of additional analysis in the revision, we believe we have made a stronger case that the neural activity we see in the dmPFC represents working memory. We acknowledge that the question of whether it is a general or less general mechanism is unaddressed (see above comments). We have revised the language as outlined above to be more focused on studying the mechanisms of persistent activity, and less focused on interpreting the significance or generalizability of our findings.

Reviewer #3 (Remarks to the Author):

In this interesting study the authors recorded units in different subregions of the prefrontal cortex (PFC) and directly compared their firing properties during the execution of working memory (WM) in a delayed non-match to sample paradigm. The main finding of this study is that, during WM delay, dorso-medial PFC (dmPFC) neurons maintain consistent firing patterns that contain information about which reward port served as the 'sample' in the current trial. This finding is complemented by observations on task-phase and outcome selectivity of neurons in the different regions. The findings are highly relevant to the field and are presented in a very clearly written and well-structured manuscript. Moreover, the data are clearly presented in the figures, which allows the reader to easily grasp the main findings.

While I fully support publication of this manuscript, I think that a number of minor (and mostly technical) points should be addressed to further strengthen the conclusions from this study:

1. SVM decoding (line 106, Fig. 2): This analysis could be further strengthened by comparing the decoder results to shuffled data (e.g., by using surrogate data for each pseudopopulation composed of spike trains for which the inter-spike intervals have been shuffled). This way, a proper statistical analysis could be performed.

Reply: Thank you for this suggestion, it greatly improves this analysis. We have updated all of the initial cross-temporal SVMs in Figs. 2g, 3g, and 4g to display significant decoding compared to a shuffled control. The shuffling was done by randomly permuting the sample port identities across trials on individual sessions.

Moreover, it would be informative to see whether the decoding results hold if performed for each mouse separately. With that approach, fewer neurons would be available for the model, so we would expect overall lower decoding accuracy, but the authors could still evaluate those results against shuffled data.

Reply: This is also a good idea. Another reviewer suggested that we actually go one step further and partition this data into individual sessions rather than animals. Therefore, we added three new supplementary figures (5, 6, and 7) using session-based cross-temporal SVMs to look at sample identity decoding capability during the delay (similar SVMs to Fig. 3g). For your reference, we have added the number of neurons recorded in each session at the top of every subpanel.

We report several interesting results from this analysis. The most relevant finding to the current comment was the fact that in the dmPFC, 5 out of the 6 mice had at least one session with substantial cross-temporal decoding of sample identity throughout the delay. Three of the mice also had more than one session with strong and stable decoding. Moreover, the one mouse (dmPFC 4) which did not have any sessions with cross-temporal decoding was the same mouse we identified in the supplemental behavioral analysis (Supp Fig. 1) to possibly not be doing working memory. This finding provides strong evidence that most of the mice are doing working memory, and that the neural activity we see during the delay is related to working memory processes in the medial prefrontal cortex. The other two subregions did not seem to contain any sessions from which we could retrospectively decode the sample port identity.

Finally, a larger number of MO neurons shows significant discrimination between the two sampling ports. Does that mean that significant decoding can be achieved with fewer neurons from that compared to the other regions? This could be tested by randomly subsampling from the pseudopopulation (or on a mouse-by-mouse basis) using increasing numbers of units.

Reply: This is an interesting thought. We added a plot of this in Supplementary Fig. 3. It looks like you are correct in that the MOs can decode better with fewer neurons, at least in the on-diagonal (trained and tested on the same time bin) time points. It was much worse in the off-diagonal points (trained on one time bin and tested on neighboring ones), indicating that overall, more neurons exhibit selectivity over shorter periods in the MOs.

2. Figs 3 and 4 show differential dmPFC activity during the delay and prior to the choice. Since information about the sampling port visited earlier in the trial is the WM component of this task, this provides evidence for a persistent activity mechanism. However, the authors rightfully mention (in line 130) that a 'temporal code' has been proposed as an alternative mechanism to persistent activity. To strengthen the authors claim that persistent activity is the dominant mechanism in this task, it would be very interesting to thoroughly check the data for traces of temporal coding. For instance, would it be possible to decode trial outcome (or sampling port identity) during the delay based on temporal information only (e.g., from the time of peak activity of each neuron)? Or, when sorted by time of peak activity during the first delay period, do the neurons fire again in the same order during subsequent delays of the same trial type? The latter could be tested for significance based on correlations of the obtained temporal tuning functions compared to those of shuffled data.

Reply: This is another interesting suggestion. We ran a delay SVM analysis similar to the one performed in Fig. 3g, except instead of using firing rate in each time bin, we collapsed across time by taking the bin number with the maximum firing rate for each neuron on each trial for left and right sample pokes, converting the data into a temporal code. We then ran an SVM 10 separate times for each region's subsampled, port-separated trials (similar to the procedure described in the methods), to attempt to decode sample port based on the timing of the peak

firing rate during the delay. With the full pseudopopulations, we found a significant main effect of subregion on temporal coding using a one-way ANOVA across the 10 subsamples ($F(2,27) = 6.12, p = .0064$). Bonferroni post-hoc comparisons revealed only a significant difference between MOs and dmPFC and no difference between dmPFC and vmPFC. With this initial analysis, we might have concluded that there does seem to be a temporal code present in the dmPFC that can differentiate left versus right sample ports.

However, we were concerned that the observed difference might be due to contamination from the small group of persistently selective neurons. Therefore, we removed the neurons previously found to be persistently selective (see Supp Fig. 4) and ran the analysis again. After their removal, there was no significant main effect of subregion on temporal coding ($F(2,27) = 1.29, p = .29$), indicating that there does not appear to be a temporal code and that the persistent firing rate code is the main form of retrospective sample identity selectivity in the dmPFC. We have added a small explanation about this to the manuscript section **“Retrospective sample port information is stably maintained in the dmPFC throughout the delay period”**.

3. On a more conceptual level: Fig. 3G shows that the decoding accuracy of dmPFC neurons drops substantially later in the delay, even when only considering the same training/testing time bins. Does this mean that information about which sample side has been visited earlier in the trial gradually disappears from the spiking of these neurons? If so, I'm puzzled how the information comes back, since Fig. 4G shows very high accuracy prior to the choice (when the animals already approach the choice port, I assume). It would be interesting to provide an interpretation of that observation in the discussion.

Reply: We agree that this is an interesting observation, however, when we switched to the 200 ms time windows for these cross-temporal analyses in Fig. 3g, the late delay decoding does not drop nearly as much as before. However, this observation is still relevant to the representational stability analysis around subsequent pokes in Fig. 7. This type of phenomenon could possibly arise due to “activity silent: working memory ([https://www.cell.com/trends/cognitive-sciences/fulltext/S1364-6613\(15\)00102-3](https://www.cell.com/trends/cognitive-sciences/fulltext/S1364-6613(15)00102-3)), which is where information is stored in the transiently increased synaptic weights of specific patterns of synapses and can be reactivated by a familiar input. In relation to your above question, we think there are likely several types of working memory occurring simultaneously as animals perform a given task, but it is beyond the scope of this current study to tease them apart in detail. Furthermore, it is also quite difficult to study activity silent working memory, and in our current dataset it is not possible to confirm whether or not activity-silent is a mechanism behind this observation. We have added a sentence about this in the results subsection **“GLM beta weights for retrospective sample location in dmPFC confirm its representational stability”** pertaining to Fig. 7.

4. Fig. 5C (line 166). The multi-class SVM is very accurate with neurons of all three regions. Given that more of the MO cells' variance in firing is explained by poke context (Fig. 5A), I wonder whether the relatively good performance of the other regions might be a saturation effect due to the number of neurons used for decoding. Again, analyzing accuracy as a function of randomly drawn neurons could help to identify more subtle differences between the regions.

Reply: This is a great suggestion and we have added it to the main figure as Fig. 5d. It looks like it confirms that the MOs need fewer neurons to encode the Poke Context.

5. Anatomy of recording sites: Would it be possible to state in which anatomical PFC subregions (e.g., piriform cortex, cingulate cortex) the individual recording sites are located? From the histology shown in Fig. 1 it is not clear whether all recording sites from each experimental group (e.g., dmPFC) fall within the same 'classical' anatomical region.

Reply: In this case, we wanted to describe the subregions in more general terms, since the locations in the paper are only the final locations after several electrode advancements and we could not definitively say where every single recording from that animal exactly came from. Also, these subregions are not consistently defined, with borders that vary depending on which mouse brain atlas one uses. For example, in our task there are 3 dmPFC mice which look like they may be in the anterior cingulate cortex, and the 3 others around the border between anterior cingulate and the medial supplementary motor area. This makes it difficult to firmly establish where precisely this persistent activity predominates when thinking about these subregions.

6. Pseudopopulation sizes: The authors state in line 80 that the pseudopopulations for the three regions differed in size. Did they control for this (by subsampling) in the SVM analyses? This should be clarified.

Reply: We did not, but for the majority of our analyses we tried to use and run statistics on the *proportions* of neurons changing their activity rather than raw numbers to hopefully account for different population sizes. The shuffling procedures to determine significance would also control for the differing numbers of neurons. We have also added the analyses requested around using different numbers of neurons for decoding which are related to the concerns expressed.

7. The task design used by the authors is quite elegant, and the fact the the rewarded port is not known until after the delay is a clear strength of this task. However, I wonder whether trials with the two possible target sites differ from each other in terms of difficulty. Given the left port is the sample, as shown in the example in Fig. 1A, would a trial in which 'center' is the target be more difficult than a trial with 'right' as the target? After all, the non-match and sample sites are closer together in that case. Would it be possible to check the performance for both near and distant match sites separately to rule this out?

Reply: The other reviewers also had similar questions. Thus, we added Supplementary Fig. 1 with some additional behavioral analysis. This shows that the center port is actually easier for the mice (Supp Fig, 1a), possibly because the center port choice trials occur 50% of the time and left or right choices occur 25% of the time. This means that the mice have twice as much practice on center choice trials. In the rest of Supplementary Fig. 1, we argue that the mice are largely not treating the center port differently than outer ports and that the pattern of behavior is suggestive of mice that are indeed using WM to perform the task.

Reviewers' comments:

Reviewer #1 (Remarks to the Author):

The authors have taken an enormous effort to further improve their excellent manuscript and to provide detailed arguments for their methodical approach and empirical observations. I have no remaining concerns or suggestions for further improvement.

I look forward to see it published in Communications Biology.

Dennis Kätzel

Reviewer #2 (Remarks to the Author):

The authors have addressed many of my and the other reviewers' concerns through new analyses, additional figures, and revisions to the text, which have significantly improved the manuscript. I have only a few remaining comments, and I have maintained the previous numbering for consistency.

- 2/3: The authors conducted decoding analyses on individual sessions and present some new findings. In their response, the authors noted, "The most relevant finding to the current and the following reviewer comments was the fact that in the dmPFC, 5 out of the 6 mice had at least one session with substantial cross-temporal decoding of sample identity throughout the delay." One animal appears to be using an alternative strategy to solve the task, which is interesting and well-described in the manuscript. However, what could explain the absence of a signal in the sessions of the other animals? The abstract states, "Dorsomedial PFC contains a stable population code, including persistent sample-location-specific firing during the delay period." However, the finding that only "at least one session" per animal exhibited substantial decoding is less robust than expected. Do the authors attribute this to technical limitations or to variations in the animals' strategies? At first glance, decodability does not seem to correlate with the number of cells recorded. Does the decoding performance correlate with the animals' task performance? To explore this, the authors could plot decoder performance during the delay for each session against parameters such as the number of cells recorded/used for decoding, session performance, trial type, and time between sample and choice.

Given that no simultaneous recordings were conducted in the different brain areas investigated, and thus no direct comparison of signal strength can be made, it is important

to consider these factors also in sessions from other brain areas. For instance, might there be stronger representations of working memory in other regions when the animal performed particularly well or when a large number of neurons were recorded simultaneously in that session?

- 4 please also state the range of # neurons recorded in each session (min - max)
- 7 last section: the decoder in the main figure with the pseudopopulation reaches almost 100%, in the plot shown here where performance is plotted against time to delay it appears much worse – why? The author could directly test the effect of number of recorded cells for instance.
- 8 the legend says for a ‘specific sample location’ - is this the actual (real) sample location or can it also be a representation of the other sample location in any given trial?

Reviewer #3 (Remarks to the Author):

The authors have done a great job in addressing the points raised by all reviewers.

Dear Editor and Reviewers,

Thank you again to the reviewers for their thoughtful comments which helped create a much more complete and compelling manuscript. We have addressed Reviewer #2's additional comments below. In addressing the comments, we generated additional data which was sufficiently interesting to add to the manuscript as an additional supplemental figure. Since there were no issues with the red text added to the manuscript from the first revision, we have changed this text to black and the remaining red text reflects new changes unique to this second revision. Similar to the first revision, the reviewer comments are highlighted in green, and our responses are shown in blue text.

Reviewers' comments:

Reviewer #1 (Remarks to the Author):

The authors have taken an enormous effort to further improve their excellent manuscript and to provide detailed arguments for their methodical approach and empirical observations. I have no remaining concerns or suggestions for further improvement.

I look forward to see it published in Communications Biology.

Reviewer #2 (Remarks to the Author):

The authors have addressed many of my and the other reviewers' concerns through new analyses, additional figures, and revisions to the text, which have significantly improved the manuscript. I have only a few remaining comments, and I have maintained the previous numbering for consistency.

• 2/3: The authors conducted decoding analyses on individual sessions and present some new findings. In their response, the authors noted, "The most relevant finding to the current and the following reviewer comments was the fact that in the dmPFC, 5 out of the 6 mice had at least one session with substantial cross-temporal decoding of sample identity throughout the delay." One animal appears to be using an alternative strategy to solve the task, which is interesting and well-described in the manuscript. However, what could explain the absence of a signal in the sessions of the other animals? The abstract states, "Dorsomedial PFC contains a stable population code, including persistent sample-location-specific firing during the delay period." However, the finding that only "at least one session" per animal exhibited substantial decoding is less robust than expected. Do the authors attribute this to technical limitations or to variations in the animals' strategies?

Reply: We think that there is most likely a technical explanation for this, which has to do with the fact that we are advancing our electrode bundle and recording new neurons in every session. The delay decoding is likely heavily reliant on persistent neurons, which only make up about 9% of all recorded dmPFC neurons (Supplementary Fig. 4). Since this is a relatively small proportion, this means that on sessions with lower neuron counts, we may, due to typical sample variation, not record any of these persistent neurons. This would account for our observation that only some sessions exhibit high and stable decoding. Indeed, there are sessions with few neurons in which we see strong persistent decoding (e.g. dmPFC 2, Session

2), and some sessions with many neurons with low decodability (e.g. dmPFC 6, Session 3). One of the drawbacks of using SVMs this way is that, although they are technically taking the entire population into consideration when decoding, they can be very sensitive to the differences in the preferred vs non-preferred firing of a small proportion of neurons, achieving high decoding accuracy from that small proportion.

At first glance, decodability does not seem to correlate with the number of cells recorded. Does the decoding performance correlate with the animals' task performance? To explore this, the authors could plot decoder performance during the delay for each session against parameters such as the number of cells recorded/used for decoding, session performance, trial type, and time between sample and choice. Given that no simultaneous recordings were conducted in the different brain areas investigated, and thus no direct comparison of signal strength can be made, it is important to consider these factors also in sessions from other brain areas. For instance, might there be stronger representations of working memory in other regions when the animal performed particularly well or when a large number of neurons were recorded simultaneously in that session?

Reply: Thank you for this suggestion; it yielded some interesting results that we have added as Supplementary Fig. 8 and attached below. We also added a couple of sentences about this finding in the main text (line 200). We found that the number of cells recorded, session performance, and time to choice correlated with decoder performance in dmPFC but not MOs and vmPFC (contrast middle column, dmPFC vs the other two columns). These findings strengthen and reiterate our conclusion that the dmPFC is the only one of the three regions studied that stably represents working memories.

Supplementary Fig. 8: Session-based correlations between mean delay cross-temporal SVM decoding strength and three behavioral parameters. The mean SVM decoding performance during the delay for each session was plotted against neuron count per session (top row), overall session performance (middle row), and time to choice poke from the end of the delay (bottom row). The correlation strength and significance are plotted above each comparison. The only significant correlations are found in the dmPFC.

• 4 please also state the range of # neurons recorded in each session (min - max)

Reply: This information is located in Supplementary Figures 5-7 at the top of each panel.

• 7 last section: the decoder in the main figure with the pseudopopulation reaches almost 100%,

in the plot shown here where performance is plotted against time to delay it appears much worse – why? The author could directly test the effect of number of recorded cells for instance.

Reply: We apologize for not providing a more detailed methodology in the earlier response. The reason for this discrepancy is two-fold. The first is that we had much fewer neurons in individual sessions than in the pseudopopulations, which will typically lead to lower decoding (e.g. see Fig. 5d and Supp Fig. 3 where we showed increasing mean decoding with neuron number). The second related reason has to do with the way we calculated the mean decoding for this figure (figure shown again below for clarity, each dot is a session). The mean decoder performance was found by taking the mean of the entire ~3 second time box from the sample to the approximate delay poke. This method captures both on- and off-diagonal decoding. In the main Fig. 2g, only the strong on-diagonal decoding is shown, and the weaker decoding is covered in grey squares, which may lead one to believe the decoding in the whole 3 second time box is higher, when it is not. Indeed, off-diagonal decoding is generally lower even for the pseudopopulation in Supp Fig. 3. So, for both of these reasons, we see lower decoding in individual sessions.

• 8 the legend says for a 'specific sample location' - is this the actual (real) sample location or can it also be a representation of the other sample location in any given trial?

Reply: We have updated the wording in the legend for Figs. 2, 3, and 4 to make it more clear that we are calculating the percentage of neurons selective for *either* sample location (essentially combining left-preferring and right-preferring neurons).

Reviewer #3 (Remarks to the Author):

The authors have done a great job in addressing the points raised by all reviewers.

REVIEWERS' COMMENTS:

Reviewer #2 (Remarks to the Author):

By clarifying the manuscript's methodology and providing some more analyses the authors have addressed all my remaining concerns.